

# Impact of spatial resolution on large-scale ice cover modelling of mountainous regions

Helen Werner[1, 2], Dirk Scherler[1, 3], Tancrède P. M. Leger[4], Guillaume Jouvet[4], Ricarda Winkelmann[2, 5, 6]

[1]Organic and Earth Surface Geochemistry, GFZ Helmholtz Centre for Geosciences, 14473 Potsdam, Germany
[2]Earth Resilience Science Unit, PIK Potsdam Institute for Climate Impact Research, 14473 Potsdam, Germany
[3]Institute of Geographical Sciences, Freie Universität Berlin, 12249 Berlin, Germany
[4]IDYST, Faculty of Geosciences and Environment, Université de Lausanne, CH-1015 Lausanne, Switzerland
[5]Institute of Physics and Astronomy, University of Potsdam, 14476 Potsdam, Germany
[6]Department of Integrative Earth System Science, Max Planck Institute of Geoanthropology, 07745 Jena, Germany

*Correspondence to*: Helen Werner (helen.werner@gfz.de)

**Abstract.** Modelling the response of mountain glaciers to anthropogenic or paleo climate change provides valuable insights given their influence on landscapes and water resources. To compensate for the high computational costs when modelling large-scale glaciers, ice fields or ice-sheets over multiple millennia, it is common practice to coarsen the spatial resolution of numerical models to typically 1–20 km, which is not sufficient to describe complex valley topographies. In this paper, we examine the

influence of spatial resolution by modelling a growing and retreating ice field at resolutions ranging from 50 m to 2 km using the Instructed Glacier Model (IGM). We find that while ice-covered areas remain similar at all resolutions, ice thickness, flow, and thermal regimes vary non-linearly with altitude in three resolution modes. The highest sensitivity to resolution is characterized by particularly strong changes in simulations within the critical mode at ~400–800 m resolution. At finer resolutions, ice flow is more topographically constrained, resulting in consistently faster flowing and thinner glaciers. In contrast, topographic resampling to

coarse resolutions lowers slope angles as well as mountain peaks and raises valley floors, supporting ice growth across all altitudes and prolongating glacial response times. Slower temperature change partially reduces the hysteresis between climate forcing and glacial response but has limited impact on resolution effects. Identifying the critical mode of strong resolution sensitivity is essential, as seemingly stable model results at coarse resolution may be misleading and accurate glacier geometries might arise from parameter choices that compensate for poorly resolved topography. We expect similar non-linear and altitudinal-

dependent resolution effects in mountain regions worldwide and emphasize the need for model advances to enable simulations at sufficiently high spatial resolutions to accurately resolve glacier dynamics.

## 1 Introduction

Mountain glaciers are a crucial component of the Earth's cryosphere, supplying freshwater for approximately 1.9 billion people (Immerzeel et al., 2020). Anthropogenic climate change threatens these resources and simultaneously increases the risk of natural

hazards in mountainous regions, including ice-rock avalanches and glacier lake outburst floods (Huggel, 2018; Hartmeyer et al., 2020). Globally, glacier melt contributed to about 20 % of sea-level rise between 2000 and 2019 (Hugonnet et al., 2021). During the Quaternary period, many mountain ranges have been extensively glaciated, which resulted in extensive modifications of the underlying landscape (Ivy-Ochs, 2015; Liebl et al., 2021). Steep and narrow peaks as well as over-deepened and U-shaped valleys are the legacy of these glaciations in Alpine regions (Penck and Brückner, 1909; Ivy-Ochs, 2015). To predict the impact of

climate change on present-day glaciated areas, studies of past glaciers and ice fields are crucial for understanding long-term trends and feedback mechanisms. Consequently, spatially distributed and accurate models of mountain glaciers and ice fields are essential for both future projections and paleo-reconstructions.

Alpine regions are characterized by a complex topography consisting of high mountain peaks and narrow valleys connected by

steep slopes. These topographical features influence the formation, dynamics, and mass balance of glaciers. For example, surface




elevation controls the mass balance mainly through the temperature lapse-rate, whereas the steepness of the bed controls glacier flow (Cuffey and Paterson, 2010; Egholm et al., 2011). However, the topographic details about peaks and valleys are partially lost when resampling a digital elevation model (DEM) to coarser resolution due to spatial averaging (Fig. 1d, e). For example, with 4806 m a.s.l., the Mont Blanc, the highest peak of the European Alps attains a height of 4745 m in a 100-m DEM, and only 4674

m when resampled to 1 km resolution. This leads to an overall loss of area at high altitudes in coarser-resolution DEMs (Fig. 1b). Similarly, steep slopes in high-relief mountain regions of typically at least 35 ° (e.g., Duncan et al., 2003; Korup et al., 2005) are reduced by resampling DEMs to coarser resolutions, thereby changing first-order topographical characteristics. Specifically, at elevations higher than 1000 m a.s.l., slope angles in a 100-m DEM are ~5 ° steeper compared to a 500-m DEM and even ~10 ° steeper compared to a 1000-m DEM (Fig. 1c). An accurate representation of the topography is therefore crucial for modelling the

ice flow in mountain regions. Yet, previous studies that modelled mountain-range scale glacier systems and ice fields in, e.g., the European Alps, the Tibetan Plateau, the Southern Alps of Zealand, typically used a km-scale resolution (e.g., Mey et al., 2016; Seguinot et al., 2018; Jouvet et al., 2023; Golledge et al., 2012; Zhang et al., 2022). To which extent spatial resolution affects glacial models in regions that are substantially controlled by their bedrock topography is not well known.

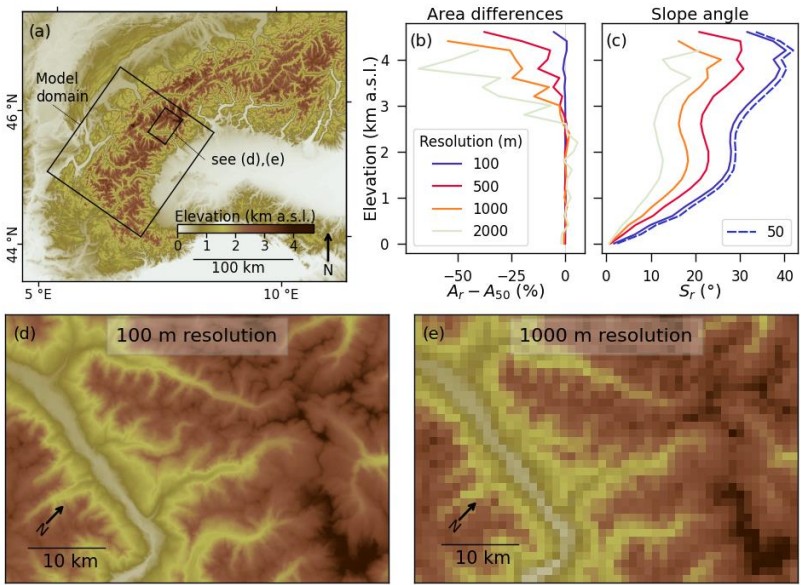

**Figure 1** Comparison of a 100 m and a 1 km digital elevation model (DEM). (a) DEM of the Western Alps at 300 m resolution (cubically resampled from a 30-m-resolution DEM from Tadono et al. [2014] with present-day glaciers and lakes removed using data from Cook et al., [2023]). Large rectangle indicates the model domain, small rectangle indicates subsets shown in (d) and (e). (b) Area differences as a function of elevation between DEMs of different resolution ($A_r$) relative to the 50 m DEM ($A_{50}$). Area corresponds to the Western Alps shown in (a) and differences are averaged in 200 m elevation bins defined by the 50 m DEM. (c) Slope angles of the Western Alps shown in (a), at 50 (blue,

dashed), 100 (blue), 500 (red), 1000 (orange), and 2000 m (grey) resolution, averaged in 200-m elevation bins defined by 50 m DEM. (d) DEM of small rectangle subset in (a) at 100 m resolution and (e) at 1 km resolution.

To improve glacier model accuracy, models are calibrated with geological data, such as mapped and dated moraines or trimlines (e.g., Kamleitner et al., 2022; Wirsig et al., 2016). However, a systematic offset has been observed between modelled and reconstructed maximum ice thickness, with models often overestimating ice thickness. In the European Alps, comparison of

modelled ice surface elevations with trimline suggests ice thickness exaggerated by 400–1000 m during the Last Glacial Maximum (LGM, ~24,000 years ago) in models based on a km-scale resolution (Seguinot et al., 2018; Jouvet et al., 2023). A better match between modelled ice surface elevations and trimlines was achieved by Mey et al. (2016), only by adjusting the sliding coefficient. Recent work by Leger et al. (2025) significantly reduced the ice thickness offset to ~150 m by using a higher



spatial resolution of 300 m in an ensemble with 100 parameter-perturbed members. However, it remains unclear how spatial
resolution influences other aspects of glacier modelling in alpine regions, and whether even finer resolutions lead to more accurate
results.

In this study, we explore the impact of spatial resolution on glacier modelling accuracy. We employ the Instructed Glacier Model
(IGM) to build a large-scale ice field over several millennia of model years, at different spatial resolutions ranging from 50 to
2000 m. Such computationally expensive experiments would be unfeasible with a traditional high-order glacial evolution model
(e.g., Elmer Ice; Gagliardini et al., 2013), however, IGM allows us to overcome this computational bottleneck by using deep-
learning to account for 3D high-order ice flow physics. In the following, we first describe the glacier model and experiment
design. Then, we present the model results, focusing on systematic differences across the simulations at different spatial
resolutions. Our analysis examines the variations of ice volume, thickness as well as other key variables with respect to spatial
distributions, temporal evolution, and bedrock altitude. Finally, we discuss the underlying processes driving the spatiotemporal
differences in ice cover and dynamics.

## 2 Methods and Materials

### 2.1 The Instructed Glacier Model (IGM)

IGM is an open-source Python model that simulates glacier evolution in 3D (Jouvet and Cordonnier, 2023). The ice flow model is
based on physics-informed deep learning which results in high computational efficiency - especially with Graphics Processing
Units (GPU). Within run times of 9 days, we were able to apply IGM to our model domain at 50 m resolution with 16,150,000
pixels over a model time of 5000 years. Jouvet and Cordonnier (2023) showed that by directly enforcing physical laws in the
learning process, IGM reproduces the solutions of a high-order analytical solver of ice flow with high fidelity. The physics-
informed deep learning approach is independent of training data from other models and thereby on predetermined spatial
resolutions. This allows us to run the same experimental set-up at different spatial resolutions and directly compare the model
results. Our IGM set-up is based on a data-consistent model of the LGM by Leger et al., (2025), using the parameterisation from
the best-scoring simulation (number 37) out of an ensemble of 100 Alps-wide simulations. This set-up is therefore well-adopted to
the Alps and the formation of an extensive ice field. The submodules describe essential glacier processes such as ice flow and
surface mass balance as well as additional modules such as avalanching or bed deformation. All model parameters can be found in
Table S1. In the following, we briefly describe the IGM-submodules and set-up.

The ice flow module in IGM is designed to efficiently simulate 3D velocities using a physics-informed deep learning approach.
Instead of solving costly Stokes equations or approximations thereof, ice velocities are simulated as energy-minimizing solutions
of the high-order Blatter-Pattyn model (Blatter, 1995) by an inexpensive convolutional neural network (CNN) (see Jouvet and
Cordonnier, 2023 for a detailed description). The generic emulator is capable of handling a variety of ice flow and temperature
regimes which is important for modelling ice fields since mountain glaciers exhibit a range of flow velocities (Millan et al., 2022;
Jouvet and Cordonnier, 2023). Ice flow at the ice-bedrock boundary is described by a non-linear Weertman friction condition
(e.g., Schoof and Hewitt, 2013). We used a pretrained CNN that was trained over a diverse catalogue of glaciers and flow
regimes. More specifically, the emulator was trained at 100 m spatial resolution with 10 vertical layers, fine vertical discretization
close to the ice-bedrock interface, 16 CNN-layers, and 32 CNN-output filters (Jouvet and Cordonnier, 2023). For adjustments to
new ice field states attained through run time, we retrained the neural network every seventh iteration, with a typical time step



corresponding to 0.01–0.04 model years. The on-the-run retraining in the high-resolution simulations at 50 and 100 m was performed every second iteration and computations were executed on a ~7000-core Nvidia A100 GPU.

The enthalpy module follows an energy-conservative formulation by Aschwanden et al. (2012). It distinguishes between cold and temperate ice: for cold ice at temperatures below the pressure-melting point, enthalpy is proportional to pressure-adjusted temperature, while for temperate ice, enthalpy includes an additional component incorporating the creation of water content through energy transfer. The Arrhenius factor in the Blatter-Pattyn model is calculated from pressure-adjusted ice temperature via the Glen-Paterson-Budd-Lliboutry-Duval law (Cuffey and Paterson, 2010). The sliding coefficient follows the Mohr-Coulomb

sliding law (Cuffey and Paterson, 2010) and accounts for basalt melt and meltwater production when the pressure melting point is reached. At the ice surface, enthalpy is constrained by surface temperature, which is modified by our temperature forcing (see Sect. 2.2). At the glacier bed, boundary conditions depend on the temperature of the bottom ice layer and bed surface temperature, where we set the geothermal heat flux to 0.065 Wm$^{-2}$ across the entire model domain.

The surface mass balance (SMB) computation is based on a temperature-index model (e.g., Hock, 2003) to calculate the SMB from monthly near-surface air temperature and precipitation. Ice surface accumulation is equal to precipitation when air temperatures are below 0 °C and decreases to zero linearly with temperatures between 0 and 2 °C. Ablation is computed with a positive-degree-day model using the integral formulation from Calov and Greve (2005) that accounts for stochastic temperature variations. To prevent ice flow beyond our model domain, we set the SMB to -50 m/year at a distance of 5 km from the model

domain edges.

Modelling space- and time-dependent bed deflection as a response to ice loading and unloading is realized by coupling IGM with the gFlex model (Wickert, 2015) using the two-dimensional elastic thin-plate equation in Mey et al. (2016). Based on Leger et al. (2025)'s best-scoring ensemble simulation, we assume a lithospheric effective elastic thickness of 45 km and set the frequency of

iteration to 50 years with a spatial resolution of 2 km for isostatic adjustment calculations.

Avalanching impacts ice accumulation, surface elevation, as well as flow velocity and is especially relevant for modelling ice fields at high resolution where slopes are generally steeper (Fig. 1c). The avalanche module in IGM is based on Kessler et al. (2006) and redistributes modelled accumulation downslope until the glacier surface reaches an angle of repose, here set at 45 °.

This process is applied at a frequency of five years.

### 2.2 Experiment design

As we aim to investigate resolution-related ice model differences in mountainous regions, we model a region with a complex alpine topography featuring high mountain summits as well as deep and narrow valleys. The ice field we modelled is located in the Western Alps, encompassing several mountain massifs with high elevations exceeding 4000 m a.s.l. (Fig 1a). This

mountainous region provides the glacier bed for a system of large and thick valley glaciers (>5 km wide, up to ~100 km long) as well as small tributary glaciers, which merge as they flow over steep terrain with an average slope of 21.5 ° (Fig. 1a). Our model domain is naturally bounded by lowlands to the northwest and southeast and is positioned near the southwestern end of the European Alps, limiting potential ice inflow from adjacent catchments which are not fully included in the domain. To enhance computational efficiency, we rotated the model domain by 55° and thereby reduced the number of lowland pixels that do not

contribute to the ice cover, constraining the model area to a total of 40,375 km$^2$ (Fig. 1a). Our modelled ice field on this topography covers a maximum area of ~12,700 km$^2$, comparable in size to the present-day Southern Patagonian Icefield, which



extends over ~12,200 km² (Meier et al., 2018). The bedrock topography is taken from the ALOS World 3D 30 m DEM (Tadono et al., 2014), which was resampled to 50 m using cubic convolution with present-day glaciers and lakes removed based on Cook et al. (2023). The DEM is resampled to multiples of 100 m resolution between 100 and 1000 m and multiples of 200 m up to 2000 m

using cubic interpolation. Note that for all experiments, resolution in both horizontal x- and y-directions is equal. We conducted additional experiments based on smoothed DEMs that were obtained from resampling the 2000 m DEM to finer resolutions using cubic convolution (see Supplement). Every simulation starts with ice-free conditions.

To simulate an expanding and shrinking ice-cover, we applied a synthetic temperature forcing that mimics a transient cooling and

warming within the range of temperature rates that occurred during the last glacial cycle (Jouzel et al., 2007), allowing for the build-up of a large ice field like during the LGM. The temperature forcing begins with an initialization phase of 1000 years with no temperature change, followed by linear cooling over 2000 years down to -8 K relative to the starting condition. The cooling phase is immediately followed by a warming phase over another 2000 years until the entire cooling is reversed. The cooling and warming rates are -4 and +4 K per 1000 years, respectively, and the total model time is 5000 years. We carried out additional

experiments with a slower temperature cooling and warming of -2 and +2 K per 1000 years stretched out over 4000 year-long cooling and warming phases, resulting in a total model time of 9000 years. These simulations were run at 100, 500, and 1000 m spatial resolution. All simulations start with temperature and precipitation conditions similar to the present-day. Monthly temperature and precipitation values are derived from climate data averaged over 1981–2010 from a weather station in Modane, France, located at 1228 m a.s.l. within the model domain (Météo-France, 2022). To project the temperatures across the entire

model domain, we used a lapse-rate of 6 °C per 1000 m. Monthly precipitation values were uniformly multiplied by 1.6 to obtain an ice field that covers the entire mountainous part of the model domain at the time of maximum cooling. We used this simplified climate set-up as we aimed to assess the influence on resolution on a growing and melting ice field instead of reconstructing past glaciations.

**2.3 Model output analysis**

To compare simulations across all resolutions, we resampled all model inputs and outputs to 50 m using nearest-neighbour interpolation to preserve the pixel structure while avoiding artefacts from coarse-resolution resampling. We analysed differences relative to the highest-resolution simulation (50 m) which serves as our reference simulation, as well as between successive resolutions to explore how differences in model output variables (e.g., ice thickness, velocity, temperature) evolve with increasing

resolution. We analysed spatially distributed model results within a defined mask that is based on glacial drainage basins located well-inside the modelled domain at maximum ice coverage of the 50 m simulation. We thus excluded from any post-processing analysis glacier catchments that lead to ice flow reaching the domain borders to avoid biases in results from our non-mass-conserving domain boundary scheme of ice melt. We used elevation bands of 0–1200, 1200–2400, and >2400 m a.s.l. to distinguish between typical topographic features such as deeply incised valleys at low altitudes and rugged mountain summits at

high altitudes. For the purpose of this paper, we introduce the notation $V_r$ for any input or output variable $V$ at resolution $r$ (in metres) and denote bedrock altitudes with m a.s.l. (metres above sea level).





## 3 Results

### 3.1 Resolution-related effects on topography resampling

To investigate how resampling effects the bedrock topography used for modelling, we first present an analysis of the elevation
differences that arise in specific parts of the landscape. Based on the assumption that resampling to a coarser resolution affects
large and deep valleys less compared to small tributary gorges or ravines, we assessed elevation differences for valleys of distinct
Strahler order (Strahler, 1957). Resampling of DEMs to coarser resolutions results in differences in bedrock elevation, which vary
with stream order as shown in Fig. 2. Drainages and their corresponding valleys of order 3–4, situated at mid elevations, are
particularly prone to resampling. In these valleys, average elevation differences at coarse resolution (1000 m and above) exceed
140 m, and at 2000 m resolution, they surpass 250 m with a notably high variation across the differences (Fig. 2). At streams of
lower order which are mostly located at higher altitudes, bedrock elevation tend to be higher than in the 50 m DEM, though
differences extend to more negative values than at any higher stream order. In contrast, high-order streams (6–7) and thus large
valleys are less effected, and elevation differences compared to the 50 m DEM are on average well below 100 m, except for the
2000 m DEM (Fig. 2). Notably, elevation differences are strictly positive at these stream orders, indicating resampling-induced
elevated valley floors in larger valleys. At finer resolutions (100 and 300 m), resampling effects remain minimal and below 50 m,
regardless of the stream order.

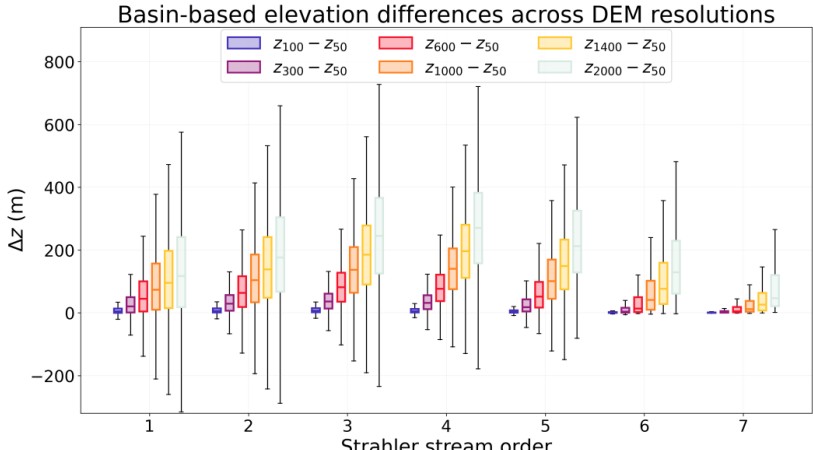

**Figure 2** Elevation differences $\Delta z = z_r\text{-}z_{50}$ in the Western Alps (Fig. 1a) between the 50 m DEM and DEMs at resolutions $r$ = 100 (blue), 300
(purple), 600 (red), 1000 (orange), 1400 (yellow), and 2000 m (grey) by valley type. The Strahler stream order (Strahler, 1957) on the x-axis
defines the stream size based on a hierarchy of tributary streams. Streams of order 1 are the smallest streams without any tributaries, usually
located at higher altitudes. As the stream order increases, the streams and associated valleys have more branches upstream and their sizes
increase. Boxes extend from the first quartile to third quartile with a line at the median. The whiskers extend from the box to the farthest data
point lying within 1.5 times the interquartile range from the box.

### 3.2 Temporal evolution of ice volume and area

In all resolution runs, the ice volume and area of the modelled ice field generally follow the temperature forcing and gradually
increase during the cooling phase (1000–3000 years), reach a maximum at the onset of the warming phase (3000–5000 years)
between 3050 to 3410 years, followed by a decrease (Fig. 3a, b). Our simulations show that a coarser spatial resolution leads to a
significant increase in ice volume, while the ice area remains largely unaffected throughout the run time. After the initialization
phase (0–1000 years), the 2000 m simulation has more than 10 times more ice volume than the 50 m run. The finest-resolution
simulations (<300 m) generate comparable maximum ice volumes of ~2000 km³ (Fig. 3a, S12). The maximum ice volume in the
intermediate-resolution simulations steadily increases with coarser resolution. The coarse-resolution simulations (>800 m)





consistently show much higher maximum ice volumes, exceeding 4000 km³ and leading to differences of a factor of ~2.5 between the 50 m and 2000 m simulations. Consequently, ice grows and melts at higher rates in the coarser resolution simulations (>800 m) with rates of ~5.7 km$^3$ per year compared to ~3 km$^3$ per year in the fine resolution runs (50–400 m) (Fig. 3a, S13).

While the overall evolution of the ice volume from the warming to the cooling phase is qualitatively the same, we observe systematic differences in the timing of maximum expansion. The maximum ice volume is reached ~200 years later in the coarser resolution (>800 m) compared to the simulations at 500 m resolution and finer (Fig. 3b, S12). The largest time lag of almost 400 years occurs between the 1000 m and the 300 and 400 m runs. For ice area, maximum values range only slightly between 11,900 to 13,000 km², and the timing offset is also smaller but still reaches up to 230 years between the finest (50 and 100 m) and

the 700 m and 1800 m resolution runs. The comparison of differences in ice volumes at given temperatures during the warming and cooling phases reveals significant hysteresis effects (Fig. 3c). At resolutions coarser than 800 m, the ice volume in the warming phase is more than 60 % higher than during the cooling phase compared to the total volume, whereas at resolutions finer than 600 m, this difference is at most ~50 %. The relative differences of ice volume between the warming and cooling phase start to increase at a temperature forcing larger than -3 K of the coarser resolution runs (> 800 m) and -3.5 K for resolutions finer than

700 m (Fig. 3c). Because the coarser resolution simulations reach the maximum ice volume relatively late after the cooling phase, the ice volume differences are highest at temperature forcing of ~-6 K which corresponds to the years 2500 and 3500, while the finer resolution simulations have their largest ice volume differences closer to the point of strongest temperature forcing.

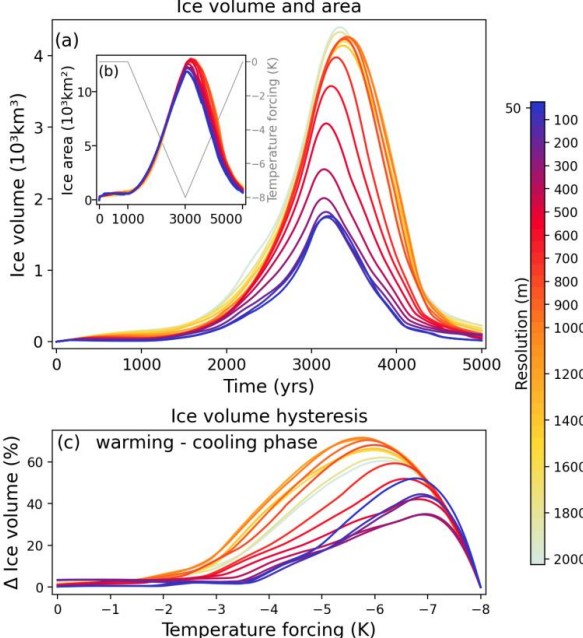

**Figure 3** Temporal evolution of (a) ice volume and (b) ice area at resolutions between 50 and 2000 m. Temperature forcing is shown in (b)
(grey). (c) Ice volume differences at times with the same temperature forcing value during the warming (3000–5000 years) and cooling phase (1000–3000 years) (warming minus cooling phase) relative to maximum ice volume. For example, a temperature forcing of -4 K is applied at model year 2000 (cooling phase) and again at year 4000 (warming phase). The relative ice volume difference at -4 K is computed as the difference in ice volume at years 4000 and 2000 relative to the total ice volume.

A distinction between low (0–1200 m), mid (1200–2400 m) and high altitudes (>2400 m) reveals pronounced differences in ice

volume, while altitudinal effects on ice area are small (Fig. 4). At the beginning of the cooling phase and end of the warming phase, the ice is mainly located at high elevations. At this time, the contribution of mid-altitude ice to the total modelled ice is small, but for ice volume (~3 %) it is twice as high compared to ice area (~1.5 %). Enhanced ice growth at mid altitudes begins



after ~1500 years of simulation and the glaciers reach lower elevations after ~1800 years. Whereas the rate of relative ice area growth at low altitudes is comparable among the resolution simulations, ice volume grows faster with finer resolution (Fig. 4b).

When the ice volumes attain maximum values, contributions to glaciated areas at mid, low, and high altitudes are about half, a third, and a sixth, respectively (Fig. 4a). The contribution to the total ice volume of low altitudes increases with finer resolution, from ~26 % in the 1000 m to ~37 % in the 100 m run (Fig. 4b). Due to the formation of thick valley glaciers, low altitudes contain more ice volume than high altitudes, although covering less ice area (Fig. 4, S11). This build-up at low altitudes likely causes the 100 and 500 m runs to show an interim relative decrease between ~2500 and 3800 years in ice volume contributions from the mid

altitudes until the ice starts to retreat. We observe a time lag when the ice melt starts, visible in both ice area and volume. The time lag corresponds to a ~500-year later decrease of the contribution at low altitudes in the 1000 compared to the 100 m simulation in both ice area and volume.

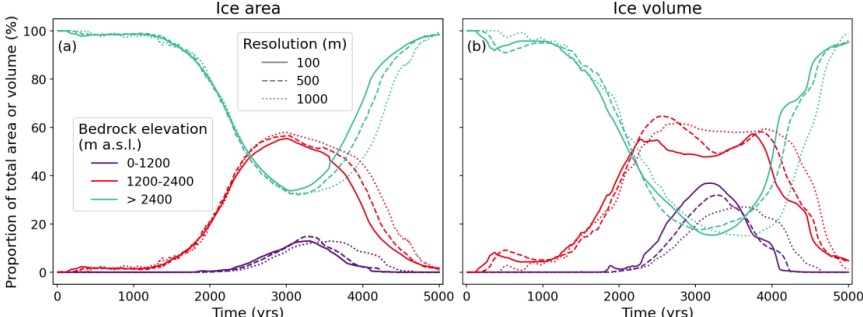

**Figure 4** Temporal evolution of the contribution of low (0–1200 m, purple), mid (1200–2400 m, red), and high (> 2400 m, green) bedrock altitudes to (a) total ice-covered area and (b) total ice volume. Results are shown for the 100 (solid lines), 500 (dashed lines), and 1000 m (dotted lines) resolution simulations.

**3.3 Effect of spatial resolution on ice field conditions at full glaciation**

When the ice volumes reach their maximum, the ice field forms a continuous ice-cover with thick valley glaciers in all resolution

simulations (Fig. S12). Figure 5 compares the maximum ice cover of each resolution run with the next coarser resolution and shows that successive ice thickness differences are smaller in finer than in coarser resolution simulations. However, spatial patterns of ice thickness differences emerge across different resolution simulations. The largest differences (in places exceeding 1000 m, both positive and negative), are found in the large valleys and at glacier termini. Below 300 m resolution, differences in ice thickness generally remain below 100 m in each pixel, except for some glacier termini, where they are related to different

extents (Figs. 5a–c, S12). At 50 and 100 m resolutions, ice is thicker in large valley glaciers and thinner in smaller tributary glaciers than in the next coarser resolution simulations. Between 300 and 700 m resolution, the coarser-resolution simulations have generally thicker ice than the finer-resolution runs (Fig. 5d–h). In contrast, finer-resolution simulations generate thicker ice at mountain tops and at some outlet glaciers. This pattern - high positive differences in large valleys and negative differences at higher elevations - is most pronounced when comparing 600 and 500 m runs. At resolutions coarser than 800 m, ice thickness

differences become significantly higher and the distinct pattern of negative and positive differences observed in finer resolution simulations mostly disappears (Fig. 5i–o). At resolutions coarser than 1000 m, both positive and negative differences are concentrated in the valleys (reaching over 1000 m), while they remain small elsewhere (Fig. 5k–o). Noticeably, the valley glacier located in the lower right of the model domain is of a very distinct piedmont-type only between 500 and 900 m resolution, despite similar glacier length at coarser resolutions (Fig. S3).





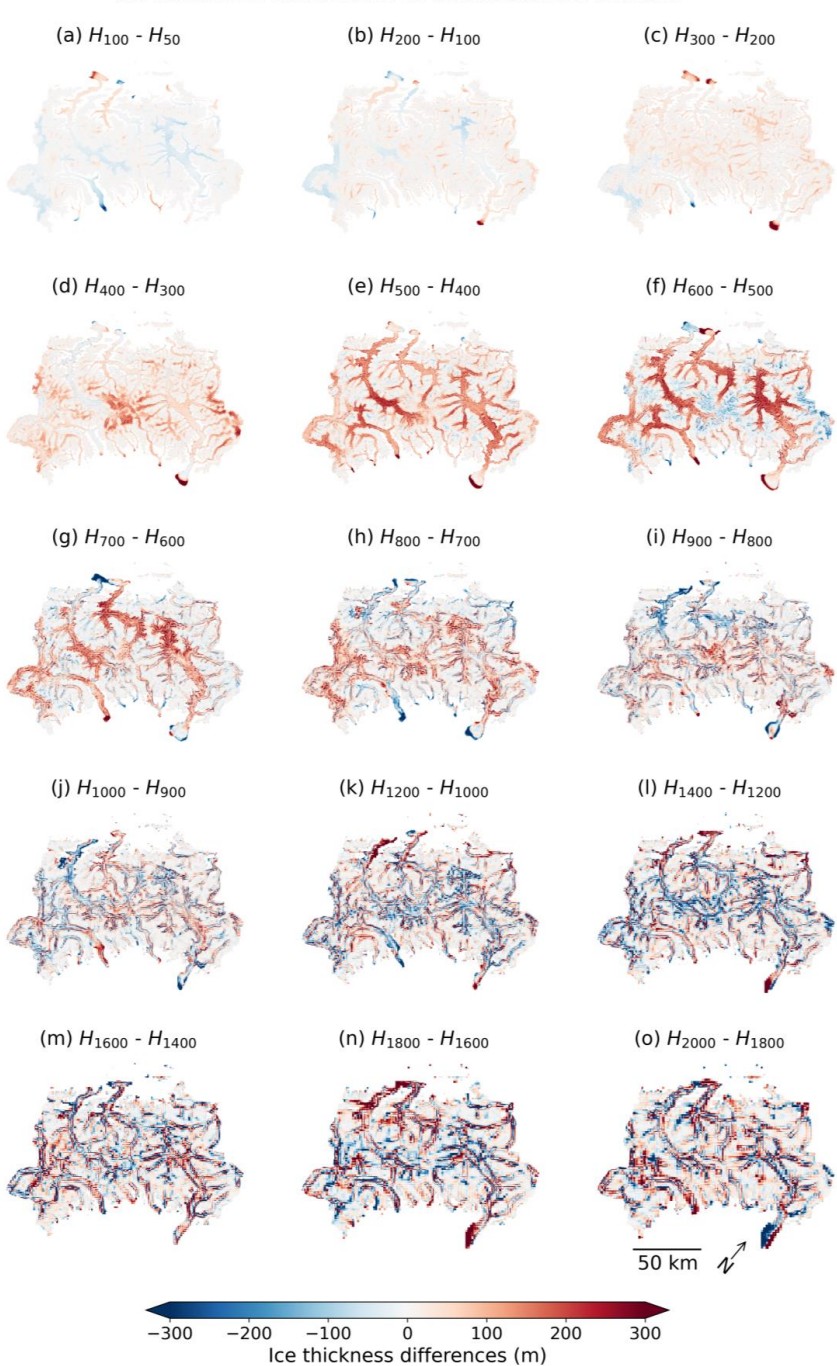

**Figure 5** Ice thickness differences ($H_{r+1}-H_r$, where $r+1$ is the next coarser spatial resolution to $r$ for $r$ = 50, 100, 200, 300, 400, 500, 600, 700, 800, 900, 1000, 1200, 1400, 1500, 1600, and 1800 m) at the time step of maximum ice volume of each resolution simulation. Blue colours indicate thicker ice at the finer resolution $r$, red colours indicate thicker ice at the coarser resolution $r+1$. In particular, this figure shows a strong thickening signal with coarser resolutions (red colours) between resolutions 300 and 700 m.






Similar to ice thickness, velocities are significantly affected by spatial resolution, especially in the main valleys (Fig. 6). At the finest resolutions (50–300 m), differences in ice velocity are generally small and ice flows slower in the finer resolution simulations than in the next coarser resolution simulation (Fig. 6a–c). In all simulations, the largest differences occur at the fast-flowing valley glaciers (Figs. 6, S4). Similar to the ice thickness, the strongest resolution sensitivity is found between 400 and 700 m resolution simulations with differences in the main valleys exceeding +600 m/year (Fig. 6d–h). Concurrently, negative

velocity differences can be observed in smaller tributaries, particularly between 400 and 500 m resolution (Fig. 6e). At resolutions coarser than 900 m, ice velocity differences remain concentrated in the valleys with a less distinct pattern and generally lower differences, remaining mostly smaller than 100 m/year (Fig. 6i–o).

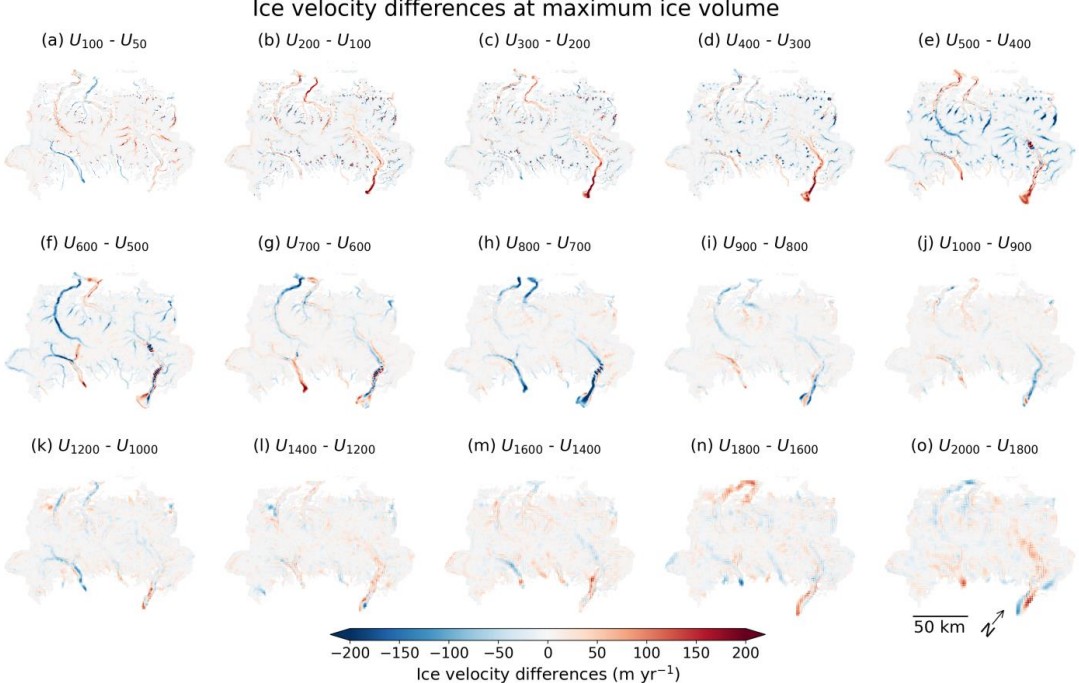

**Figure 6** Differences of depth-averaged ice velocities ($U_{r+1}-U_r$, where $r+1$ is the next coarser spatial resolution to $r$ for $r$ = 50, 100, 200, 300,
400, 500, 600, 700, 800, 900, 1000, 1200, 1400, 1500, 1600, and 1800 m) at the time step of maximum ice volume of each resolution simulation. Blue colours indicate faster ice at the finer resolution $r$, red colours indicate faster ice at the coarser resolution $r+1$.

The spatial patterns of ice thickness differences between successive resolution pairs (Fig. 5) are obscured by much larger differences when comparing to the 50 m simulation, as shown in the distribution of ice volume across bedrock elevations in Fig. 7 (see also Fig. S7). Generally, ice volume systematically increases with coarser resolution, however, we observe distinct resolution

modes: Fine-resolution simulations (≤300 m) yield similar ice volumes, as do the coarsest resolutions (≥1000 m), while the intermediate 500 m resolution simulation falls roughly halfway between these two groups, with a distinct jump in ice volume (Fig. 7a). Despite these differences, approximately 80 % of the total ice volume in all resolution runs is located between 400 and 2600 m a.s.l.. At these bedrock elevations, the volume per elevation bin remains rather constant among the finer resolutions, while volumes at coarser resolutions fluctuate by up to 50 % within the same elevation range. Compared to the 50 m simulation, the

100 m run consistently shows slightly more ice volume (Fig. 7b). At coarser resolution, however, relative differences are notably high in two elevation ranges: At bedrock elevations of ~1500 m a.s.l., where the simulations produce their largest ice volumes, the volume in both the 1000 and 2000 m runs are over 160 % higher and at 500 m resolution over 60 % higher than in the 50 m run. Secondly, at ~3200 m a.s.l., all resolution simulations have significantly more ice volume than the 50 m run (Fig. 7b). Particularly




high relative differences, exceeding 400 %, are observed above 4000 m a.s.l. due to low (<1 km³) absolute ice volumes. At pixels,

where the 50 m run holds more ice than in coarser simulations, differences remain below 20 km³ and ~20 % at all resolutions. At

the lowest elevations, the 100 and 300 m runs produce only slightly more ice volume than the 50 m run, while relative differences

at coarser resolutions are large. Absolute differences at the lowest elevations exceed 50 km³ for the 500 and 1000 m runs, and

100 km³ for the 2000 m simulation, although these values are lower than those at mid elevations.

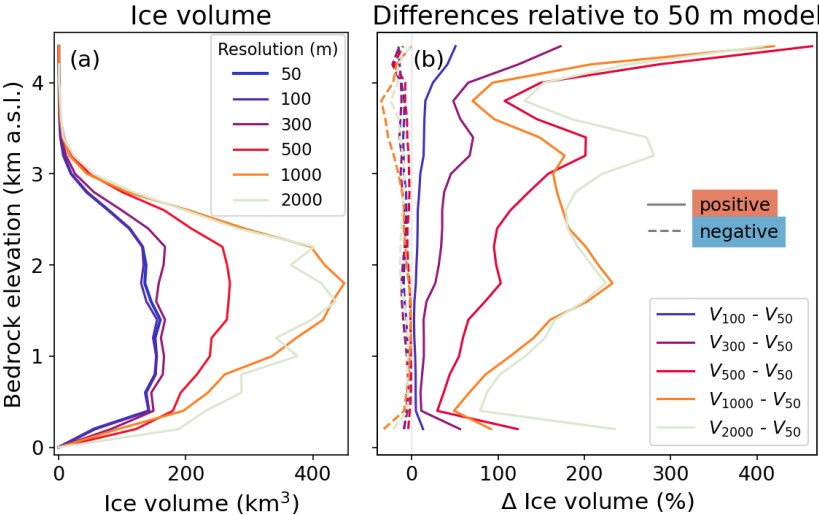

305  **Figure 7** Ice volume at the time of maximum ice volume across bedrock elevation. (a) Ice volume in 200 m elevation bins of the corresponding DEMs at 50 (thick, blue), 100 (violet), 300 (purple), 500 (red), 1000 (orange), and 2000 m (grey) resolution. (b) Ice thickness differences with reference to the 50 m simulation ($H_r$-$H_{50}$) integrated over 200 m bins of 50 m DEM bedrock elevation ($V_r$-$V_{50}$) for $r$ = 100 (violet), 300 (purple), 500 (red), 1000 (orange), and 2000 m (grey). We distinguish between positive ice volume differences where the resolution r run generates more ice (solid lines), and negative differences where the 50 m run has more ice (dashed lines).

310  During full glaciation, resolution-related differences in model variables depend on bedrock altitude and are most pronounced

between 300 and 700 m resolution (Fig. 8). While input elevation and slope vary monotonically with resolution (Fig. 8a, b), the

changes in the output variables do not reflect this pattern and are more non-linear (Fig. 8c–f). Consistent with our starting

observations (Figs. 1, 2), the 50 m DEM has up to 50 m lower elevations at low and ~20 m lower elevations at mid altitudes,

while mountain tops are up to 60 m higher compared to the 2000 m resolution DEM (Fig. 8a). Similarly, bedrock slopes are

315  gentler in the coarser resolution DEMs, especially at high altitudes where they are up to 14 ° lower (Figs. 1, 8b). Among the finest

resolutions (100–400 m), mean differences compared to the 50 m DEM are small and remain below 7 m in elevation and 5 ° in

slope. Although the modelled ice is consistently thicker at coarser resolutions, the increase is not linear: At low and mid altitudes,

mean ice thickness increases only slightly at the finest resolutions (<500 m), followed by a drastic increase between 400 and

800 m resolution - by 56 % at low altitudes and even 115% at mid altitudes - and levels off at coarser resolutions (Fig. 8c). A

320  similar pattern occurs at high altitudes, though the steep increase in ice thickness occurs at finer resolutions, between 300 and

500 m. The largest absolute mean ice thickness differences (>300 m) between the 2000 and 50 m simulations are observed at low

altitudes, while relative differences are strongest at high elevations, with a factor of ~1.5. Similar to the ice thickness, velocity

differences at low and mid altitudes are minor at the finest resolutions and most pronounced at resolutions between 400 and

800 m, with a substantial reduction in ice velocities (Fig. 8d). At low altitudes, where glaciers flow fastest, average values

decrease from a maximum of more than 200 m/year at 500 m resolution to less than 150 m/year at resolutions coarser than 700 m.

At high altitudes, average velocity drops from ~20 m/year in the 50 m simulation to roughly half of than in the 500 m run, but

increases again at coarser resolutions. Ice temperature at the glacier base shows the highest resolution-sensitivity at high altitudes,



transitioning from cold-based conditions (with a mean basal ice temperature of -4.8 °C at 50 and 100 m resolution) to temperate at resolutions coarser than 1000 m (Fig. 8e). At mid and low altitudes, basal ice is warmer at fine resolutions, with mean

temperatures shifting from ~-0.8 °C to 0 °C by 800 m resolution, while basal ice remains temperate at low altitudes across all resolutions. The mean ice softness is rather insensitive to resolution changes and differences related to bedrock altitudes are more apparent, with softer ice at lower altitudes (Fig. 8f). At high altitudes, softness increases steadily with coarser resolution, while at mid and low altitudes, a marked increase occurs at ~500 m resolution.

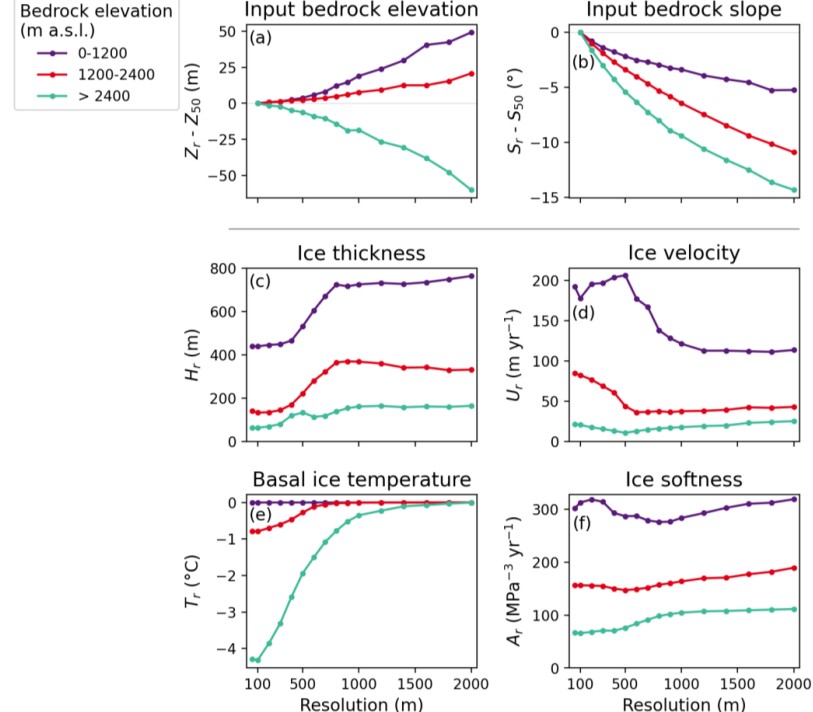

**Figure 8** Comparison of model inputs and outputs at low, (0–1200 m, purple), mid, (1200–2400 m, red), and high altitudes (> 2400 m, green). Mean differences of (a) bedrock elevation ($Z_r$-$Z_{50}$) and (b) bedrock slope ($S_r$-$S_{50}$) of the input DEMs at resolution $r$ = 100, 200, 300, 400, 500, 600, 700, 800, 900, 1000, 1200, 1400, 1600, 1800, and 2000 m compared to the highest-resolution run at 50 m. Mean variables at the time of maximum ice volume in each resolution simulation are shown in (c)–(f): (c) ice thickness ($H_r$), (d) depth-averaged ice velocity ($U_r$), (e) pressure-adjusted basal ice temperature ($T_r$), (f) depth-averaged Arrhenius factor ($A_r$) for resolutions $r$ = 50, 100, 200, 300, 400, 500, 600, 700, 800, 900,

1000, 1200, 1400, 1600, 1800, and 2000 m. All averages are taken over the glaciated area at resolution $r$ (and 50 m resolution for (a) and (b)).

**3.4 Impact of temperature forcing across bedrock altitude**

Throughout the cooling phase, ice is generally thicker and flows slower at coarser resolutions, however, non-linear resolution effects differ from the ones observed at full glaciation and the transition at ~400–800 m resolution, followed by a plateau at coarser resolutions is less consistent (Fig. 9). The largest relative ice thickness differences occur at 1500 years, when ice is

restricted to high altitudes. With coarser resolutions, thickness differences increase, and ice is twice as thick at 1000 m and four times thicker at 2000 m compared to the 50 m run. After 2000 years, glaciers extent to mid altitudes, where ice thickness remains consistent at the finest resolutions (≤200 m) and is 30-80 % higher at coarser compared to 50 m resolution (Fig. 9b, f). At high altitudes, thickness differences remain large, with over 250 % more ice at 2000 m resolution. After 2500 years, the glaciers advanced into low altitudes. Notably, this is the only time step where ice is thinner compared to the 50 m run, indicating a delayed

ice response to cooling at coarse resolutions. The spatial distribution of ice thickness differences between the 500 and 50 m simulations shows mainly positive differences, but strongly negative differences occur at smaller and tributary glaciers (Fig. 9c).



Ice thickness differences increase with coarser resolution at high and mid altitudes, but plateau more appreciably compared to earlier time steps, at ~130 % beyond 500 m (mid altitudes) and 700 m resolution (high altitudes). We observe a similar pattern of ice thickness differences at high and mid altitudes at 3000 years of model time when the ice-cover is almost at its maximum

(Figs. 9g, h, S12). At this time step, the ice thickness differences at low altitudes flip compared to 2500 years and all resolution simulations have 30-50 % thicker ice than the 50 m run. The ice velocity difference pattern across the resolutions is similar at 1500 and 2000 years: Ice flows at most 20 % slower in the 100 compared to the 50 m run and differences are more negative and rather constant at coarser resolutions (Fig. 9m, n). At high altitudes, ice flow differences relative to 50 m resolution generally become less negative at coarser resolutions towards maximum cooling (at 2500 and 3000 years), with ice flowing faster than in

the 50 m run at 3000 years of model time. The largest differences in relative (up to ~80 % slower ice flow) and absolute terms (~-155 m/year) appear at low altitudes at 2500 years and most likely reflect delayed ice growth at coarse resolution. At 3000 years and low altitudes, mean ice velocities at 200–500 m resolution are nearly identical, although the comparison of ice flow at 500 and 50 m resolution shows a notably mixed pattern of positive and negative differences that likely cancel each other out in the mean values (Fig. 9l, p). At these altitudes, the strong decrease in ice velocity differences at intermediate resolutions (500–700 m) is

followed by roughly stabilizing differences at resolutions coarser than 700 m.

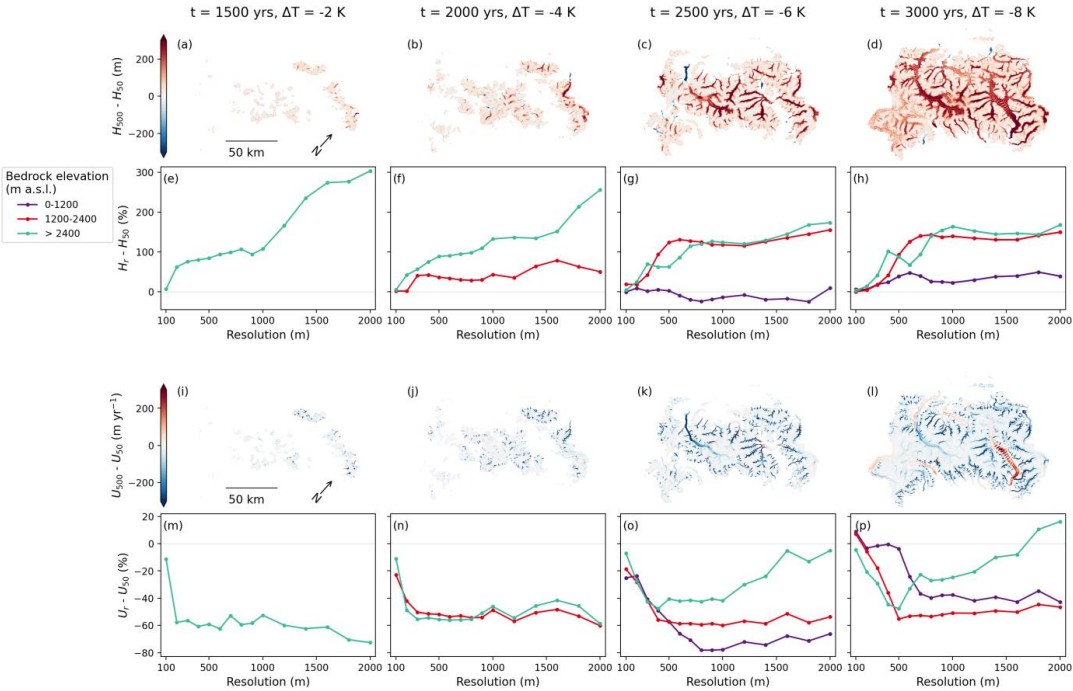

**Figure 9** Comparison of ice thickness and velocity at different resolutions to the 50 m simulation throughout the cooling phase at different time steps $t$ with corresponding temperature forcing $\Delta T$. (a)–(d) Ice thickness differences between the 500 and 50 m runs ($H_{500}$-$H_{50}$). (e)–(h) Mean differences of ice thickness ($H_r$-$H_{50}$) relative to the 50 m run. (i)–(l) Depth-averaged ice velocity differences between the 500 and 50 m runs

($U_{500}$-$U_{50}$). (m)–(p) Mean differences of depth-averaged ice velocity ($U_r$-$U_{50}$) relative to the 50 m run. Panels in the columns correspond to the same time steps: (a), (e), (i), (m) after 1500 years of model time when temperature forcing is -2 K compared to present-day, (b), (f), (j), (n) after 2000 years of model time, when the temperature forcing is -4 K, (c), (g), (k), (o) after 2500 years of model time, when the temperature forcing is -6 K, (d), (h), (l), (p) after 3000 years of model time, when the temperature forcing is -8 K. All comparisons in (e)–(h) and (i)–(l) are shown with respect to low, (0–1200 m a.s.l., dark purple), mid, (1200–2400 m a.s.l., red), and high altitudes (> 2400 m a.s.l., green) with respect to the 50 m

DEM and for resolutions $r$ = 100, 200, 300, 400, 500, 600, 700, 800, 900, 1000, 1200, 1400, 1600, 1800, and 2000 m. Averages are taken across glaciated pixels at resolution $r$ and/or in the 50 m run.



When evaluating the impact of resolution on ice thickness and velocity during the cooling phase, there is a chance that the response depends on the rate of cooling. To address this point, we compare in Fig. 10 the model results under fast (± 4 K/kyrs) and slow (± 2 K/kyrs) temperature forcing, which show that the trend of increasing ice volume with coarser resolution persists. In

fact, the resolution-related differences are more pronounced than those caused by the change in temperature forcing rate: Slower temperature forcing increases the maximum total ice volume by ~20 % compared to fast forcing. However, the differences in peak ice volume between the 50 and 2000 m simulations is much larger (138 % under fast and 162 % under slow forcing). Accordingly, the coarser resolution runs are more sensitive to changes in the rate of cooling. At 500 and 1000 m resolution, the largest volume differences between the two forcings occur at low altitudes, while the variations in the 50 m simulations are marginally larger at

mid than at low altitudes (Fig. 10b, c). At high altitudes, the absolute volume differences between the forcing simulations are minor and at 500 and 1000 m resolution, the fast forcing even yields slightly more ice than the slow forcing (Fig. 10a). Despite these differences, the distribution of ice volume across altitude bands remains largely unchanged and all simulations contain the majority of ice volume at mid altitudes. Additionally, the general pattern of ice volume change with temperature persists: Ice volume at high altitudes changes more gradually, while low-altitude regions respond with abrupt and strong increases and

decreases. The temporal delay in maximum ice volume with respect to temperature forcing is mainly reduced with the slower temperature change; however, resolution-related differences remain. In the 1000 m runs, the lag drops from 420 to 250 years, driven by continued ice growth at low altitudes under both forcings, while glaciers at mid and high altitudes are more sensitive to temperature change and start to retreat earlier. In the 50 m runs, the time lag stays near 200 years, though peak ice volume occurs later at mid than at low altitudes.

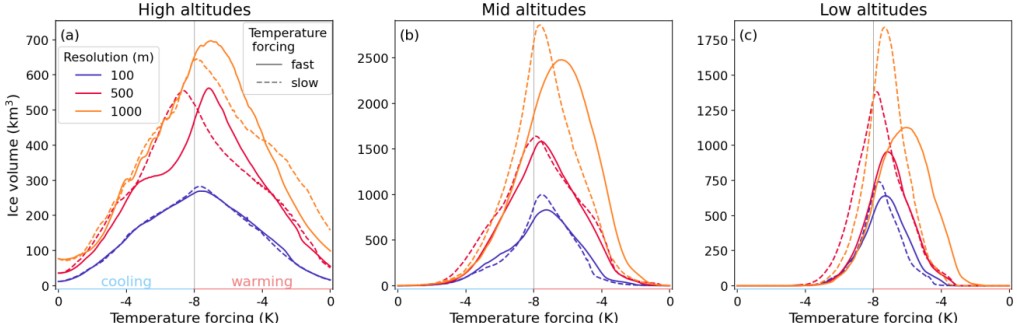


**Figure 10** Ice volume under fast (±4 K/kyrs, solid line) and slow (±2 K/kyrs, dashed line) temperature forcing at 100 (blue), 500 (red), and 1000 m (orange) spatial resolution at (a) low (0–1200 m a.s.l.), (b), mid (1200–2400 m a.s.l.), and (c) high bedrock altitudes (>2400 m a.s.l.). Temperature anomaly on x-axis starts after the initialization phase, which is 1000 years of model time under constant temperature forcing (at 0 K) and ranges from the cooling (0 to -8 K; 1000–3000 years under fast and 1000–5000 years under slow temperature forcing) to the warming

phase (-8 to 0 K; 3000–5000 years under fast and 5000–9000 years under fast temperature forcing).

## 4 Discussion

In all our simulations, we observe that coarse-resolution simulations generate more ice than fine-resolution ones, and our results suggest that the resolution-related effects are non-linear. Specifically, at the time of maximum ice volume, we identify three modes in which the changes of resolution express themselves in key variables: For resolutions coarser than 800 m, differences in

ice thickness and velocity are consistently large (Figs. 5, 7, 8). Model results of simulations at fine resolutions (finer than 400 m) are comparable, too, however, differences are generally small in magnitude and become even smaller with finer resolution (Figs. 5, 7, 8). In the critical mode at 400–800 m resolution, model variables are sensitive to even small changes of resolution and we find the highest stepwise differences in ice thickness between the 600 and 500 m runs (Fig. 5). The spatial distribution of ice thickness differences shows a distinct pattern between valleys and mountain tops, which varies by resolution mode (Fig. 5).





**4.1 Resolution-related differences across bedrock altitude**

Across bedrock altitudes, resolution-related differences mostly follow a distinct pattern. Our results suggest that ice growth is supported by the coarse-resolution topography with flattened mountain peaks and gentle slopes, while fine-resolution simulations promote ice thinning in steeper and narrower valleys.

At high altitudes, absolute differences in ice thickness and velocity are small across the resolution simulations compared to mid and low altitudes - presumably due to the generally limited area and thinner ice (Figs. 7, 8, S3, S8, S11). However, in relative terms, they are significant (Fig. 8, 9). With the accumulation areas available in the coarse-resolution topography, the gentle slopes apparently contribute to preventing glaciers from directly flowing into the ablation area and instead enhance ice build-up on flat mountain tops. The variability in bedrock elevation and slope differences among the coarsest resolutions (Fig. 8) therefore seems to lead to a wide range of ice thickness differences at the beginning of ice growth (Fig. 9). Although the coarse-resolution topography has generally less area at the highest elevations (Fig. 1), which restricts ice accumulation, this does not seem to have a strong effect on ice thickness. The increased thickness at coarser resolutions results in increasing basal pressure and thus lowering the pressure melting point, potentially leading to basal melting. Additionally, it promotes shear heating from enhanced ice deformation. These processes are expressed in higher basal temperatures at coarser resolution. The shift from frozen bed conditions at fine resolution simulations to temperate basal ice at coarser resolutions may additionally be enhanced by higher elevations in the fine-resolution DEM (Fig. 8), where colder climatic conditions prevail. The Arrhenius factor, which governs the ice softness, shows the greatest relative changes to resolution at high altitudes compared to mid and low altitudes. This presumably reflects the resolution-dependent differences in ice temperature where warmer ice is associated with higher softness, leading to faster ice flow at coarse resolutions. However, among the finest resolutions, the increased steepness of bedrock slope appears to have a stronger influence on ice velocity than a lower Arrhenius factor, ultimately resulting in faster ice flow in the 50 m simulation (Fig. 8).

At mid altitudes, the differences in thickness, velocity as well as thermal regimes between the resolution runs generally fall between those observed at high and low altitudes (Fig. 8). Generally, fine resolution simulations generate fast and thin ice, with a abrupt transition to thick and slow glaciers at coarse resolutions (Figs. 8, 9). Notably, resolution-related ice volume differences at mid altitudes are highest across the range of bedrock altitudes (Fig. 7). Valleys of intermediate stream order - commonly found at these altitudes - are especially affected by resolution changes in the DEMs (Fig. 2). In coarser resolution topographies, elevated valley floors reduce slope angles, influencing ice flow, and moreover raise ice surface elevations, which affects surface temperatures via the lapse-rate, thereby impacting glacial mass balance. These implications of DEM resampling, combined with the hypsometry of the ice field, where most of the ice is stored at mid altitudes in all resolution simulations (Figs. 7, S11), can account for the large ice volume differences at mid altitudes. Additionally, the high concentration of ice volume at mid altitudes seems to stabilize resolution-related effects, with the least variations of resolution-related patterns and most distinctive resolution modes throughout the cooling phase (Fig. 9).

At low altitudes, we observe the largest absolute differences in ice thickness and velocity across the spatial resolutions. These differences are generally negatively correlated: Finer-resolution simulations simulate thinner, faster-flowing ice, whereas coarser resolution tend to generate thicker, slower-flowing ice (Fig. 8, 9). We conjecture that stronger lateral confinement by steeper and narrower valley walls in the finer resolution DEMs forces the ice into a more constricted flow path. To satisfy mass conservation for ice as an incompressible fluid, ice flow accelerates, leading to glacier tinning. At intermediate resolutions, we observe a pronounced transition from fast and topographically constrained ice flow at fine resolutions to slower, thicker ice flowing over



smoothed topography at coarse resolution, characterized by large variations within small changes in resolution (Fig. 8). However, among the finest resolutions, ice velocity does not follow this pattern and instead decreases with finer resolution. While the shift in mean velocity from ~120 m/year at 1000 m to over 200 m/year at 500 m resolution is more pronounced than the subsequent drop to ~180 m/year in the 100 m run, the velocity reduction at resolutions finer than 500 m is noteworthy because it is observed only at low altitudes (Fig. 8). This might result from increased lateral shear stress from very steep and narrow valley walls, presumably imposing resistance and slowing down ice flow. In the valleys, the surface mass balance-elevation feedback plays an important role: When ice thickness increases, the ice surface reaches higher elevations with lower temperatures, which promotes further ice growth (Levermann and Winkelmann, 2016). This feedback is most likely strongest in the shallow-sloping valleys with steep and narrow sidewalls at fine resolution. Overall, the surface mass balance-elevation feedback seems to partly offset thinning due to acceleration from channelized ice flow, resulting in consistently thicker ice at low altitudes than at high and mid altitudes - across all resolutions (Fig. 8). At coarser resolutions, valley floors are elevated due to topographical resampling (Figs. 1, 2, 8), thereby expanding the accumulation zone and further enhancing ice build-up.

### 4.2 Resolution-dependent hysteresis effects

Delays in peak ice volume with respect to the time of maximum cooling and corresponding ice build-up during the warming phase occur in all simulations but are most pronounced at coarser resolution (up to 420 years compared to a time lag of less than 200 years at the finest resolutions) (Figs. 3, S12). Our results suggest that these hysteresis effects at full glaciation primarily originate from low-altitude regions (Fig. 4) and most likely highlight the influence of the surface mass balance-elevation feedback, which allows valley glaciers to continue thickening despite rising temperatures. Glaciers tend to be thicker, and the underlying DEM features gentler slopes at coarse resolution (Figs. 1, 8), which results in shallower surface gradients (away from glacier termini) (Figs. S3, S5, S10). Therefore, changes in mass balance affect larger areas, which increases the glacial response time compared simulations of fine resolution. A twice slower temperature forcing leads to a notable reduction of the time lag between maximum cooling and peak ice volume at coarse resolution (from 420 to 250 years in the 1000 m runs) (Fig. 10). However, the lag remains clearly evident in the coarse resolution simulations, where the slower forcing leads to a substantial ice growth at low altitudes, likely enhanced by the surface mass balance-elevation feedback (Fig. 10). For investigating more rapid climate change events in the geologic past or anthropogenic future with temperature variations of several degrees over century to millennia time scales, such as the Bølling-Allerød warming or the Younger Dryas cooling period (Rasmussen et al., 2006; Golledge et al., 2008; Norris et al., 2024; Patton et al., 2017), our results suggest that fine-resolution modelling is crucial to accurately capture the timing of glacial retreat and advances, while avoiding overestimated hysteresis effects.

### 4.3 Topographic control on resolution-related differences in glacial modelling

Gradual changes in DEMs with resolution (Figs. 1, 2, 8) cannot explain the non-linear variations in modelling results across different resolutions (Figs. 5, 6, 8, 9). Instead, our results suggest that the interplay of topographic representation and ice dynamics gives rise to three resolution-dependent modes: At fine resolutions (50–300 m), the characteristic alpine topography is well preserved. Differences in elevations and slopes compared to the finest-resolution DEM are only a few meters or degrees (Figs. 1, 2, 8) and appear to be too small to substantially affect model outputs, resulting in relatively small variations among the fine-resolution simulations. In contrast, DEMs at resolutions coarser than 800 m exhibit a qualitatively different character: While the main valley systems are retained, smaller-scale features - particularly intermediate valleys - are not accurately resolved. As a result, at coarse resolutions, the ice field is less constrained by topography and instead shows much thicker ice, slower ice flow, and higher inertia (Figs. 3, 5–9, S5). We hypothesize that at ~800 m resolution, the topographic constraints become sufficiently weakened so that further coarsening has a minor additional effect, and glacier geometry is increasingly controlled by surface



elevation in terms of ice thickness rather than bedrock relief. The critical mode of strong resolution sensitivity at 400–800 m resolution marks the transition between the fine and coarse resolution modes with their different qualities and is characterized by continuous and drastic variations with resolution.

Our results emphasize the influence of DEM resampling on bedrock elevation and slope and thus on ice modelling results, with

implications extending beyond the Western European Alps. In experiments with smoothed topographies that are on average 10 ° less steep than the present-day Alps, with mean slopes more similar to mountain ranges like, e.g., the Scottish Highlands (e.g., Whitbread et al., 2015), we find that spatial resolution has a similar influence on ice dynamics which is again dependent on bedrock altitudes (Figs. S1, S2). The critical mode in these experiments shifts to ~500–700 m resolution (Fig. S2). Therefore, the fine-resolution mode extends only to slightly coarser resolutions compared to simulations using the original DEM. This

emphasizes the importance of running simulations at resolutions finer than km- and ideally 500-m-scale, even in relatively smooth terrain. Although our model domain is located in the European Alps, similar resolution-dependent patterns are thus expected in glacier models of other mountainous regions. Additionally, assuming an inherent characteristic wavelength of landscapes (Perron et al., 2008; Voigtländer et al., 2024; Grieve et al., 2016), similar modifications of elevation, slope, and valley shapes (Figs. 1, 2, 8) are expected when resampling to resolutions coarser than the dominant wavelength. We therefore speculate that while the

specific altitude bands may shift depending on regional topographic features, thresholds of the resolution modes probably vary only slightly and non-linear resolution effects at low, mid, and high altitudes likely persist. This highlights the need for further research on resolution-related effects on different ice fields.

Our findings emphasize that modelling topographically constrained glaciers or ice fields requires careful consideration of spatial

resolution. Increasing resolution from coarse to fine can improve the agreement between geological evidence and numerical models that tend to overestimate ice thickness of past ice fields like in the European Alps (Seguinot et al., 2018; Jouvet et al., 2023), as shown by Leger et al. (2025). Moreover, this correction of ice thickness overestimations at similar ice extents leads to more realistic ice loading reconstructions of lower magnitude, which in turn refine assessments of isostatic rebound. For example, lithospheric deflection in the European Alps is estimated to be between 1.5 to 3 times lower using a 300 m DEM (Leger et al.,

2025) compared to km-scale grid (Mey et al., 2016; Jouvet et al., 2023). Crucially, such improvements depend on identifying the critical mode of strong resolution sensitivity where simulations transition markedly from a topographically controlled to a smoother and thicker ice field. Since model outputs appear stable across different resolutions among the coarse resolution mode, an apparent resolution insensitivity does not necessarily indicate that the chosen resolution is sufficient. In addition to using appropriate climate and SMB forcing and exploring model parameters, it is also important to use a fine spatial resolution that

accurately resolves the topography. A series of tests and considerations of spatial difference patterns might help to detect the critical resolution mode to ensure that a finer resolution does not substantially influence the ice dynamics - especially in smoother terrains that seems to narrow the critical mode of strong resolution sensitivity. Otherwise, the model parameters may be tuned to produce seemly accurate glacier geometries that, in fact, compensate for unrealistic terrain representations. Additionally, as glacier models further advance, fine spatial resolutions will likely become even more critical. Incorporating erosion processes

could further enhance resolution-related effects by altering valley shapes (Liebl et al., 2021; Valla et al., 2011). The accelerated ice flow in fine-resolution models is expected to stronger carve out larger trunk valleys, whereas thin and bed-frozen ice-covers shield high altitude plateaus from erosion processes, and ice-free parts are exposed to non-glacial erosion (Bernard et al., 2025; Egholm et al., 2017). Similarly, differences between ice models using full-Stokes or Blatter-Pattyn implementations might become more relevant at fine resolutions (Rückamp et al., 2022). However, even fine-resolution DEMs are only imperfect

approximations of real-world topography and surface slopes remain underestimated if the inherent projection error caused by



representing 3D surfaces on 2D grids is not corrected (Voigtländer et al., 2024). These challenges highlight the importance of continued model development to allow for higher-resolution modelling over large time and spatial scales at reasonable computational costs, for example through GPU-based modelling approaches.

## 5 Conclusions

To assess the impact of spatial resolution on model outputs, we simulated the hypothetical growth and decline of a ~12,700 km$^2$ ice field over 5000 years at 16 different resolutions ranging from 50 to 2000 m. The simulations over both, large spatial scale and several millennia were made possible by the GPU-based architecture of the Instructed Glacier Model (IGM), which allows for high computational efficiency and comprehensive comparison across fine and coarse spatial resolutions. Our analysis led us to the following findings:

i.    Coarser resolution simulations generate much thicker ice and slower ice flow while keeping a similar glaciated area.

    ii.    Resolution-related differences in the modelled ice field dynamics are non-linear. We find three distinct modes of influence, with strong sensitivity in model outputs from ~400 to 800 m resolution, separating relatively consistent results at both finer and coarser resolutions. Identifying this critical mode of strong resolution sensitivity is key, as seemingly stable model output at coarse resolution may be misleading and arise from ice dynamics driven by inaccurately resolved topography. At fine resolutions, bedrock slopes appear to be sufficiently well-resolved to control ice flow, resulting in a more topographically constrained ice field.

    iii.    At high altitudes, coarse-resolution simulations generate more ice due to flatter mountain tops and shallower slopes in the corresponding topography, providing a larger accumulation area and slower ice flow. The thick ice ultimately leads to warm-based glaciers at full glaciation while the thinner glaciers in the fine-resolution runs remain cold-based.

iv.    In the valleys, the deeply incised topography leads to ice flow speed-up in the fine-resolution simulations, resulting in ice thinning while the elevated valley floors in the coarse-resolution runs enhance ice thickness increase.

    v.    Intermediate valleys at mid altitudes raised by DEM resampling cause the largest resolution-related ice volume differences across bedrock altitudes due to the glacial hypsometry.

    vi.    Thicker ice in the coarse resolution simulations is associated with shallower surface slopes which prolong glacial response time. Slower temperature cooling can reduce the time lag and hysteresis between temperature forcing and glacier evolution; however, the resolution-related variations at bedrock altitudes remain.

    vii.    Experiments using a smoothed DEM suggest that the non-linear, altitudinal-dependent influence of spatial resolution might be expected across other mountain regions notably less steep than the European Alps as well. The critical mode of strong resolution sensitivity may appear in a narrower range and shift to only slightly coarser resolutions, highlighting the need for further development of models capable of high-resolution simulations.

## 6 Code and data availability

The Instructed Glacier Model (IGM) source code (Python programming language) and documentation are available from Guillaume Jouvet's GitHub repository at https://github.com/jouvetg/igm. We used IGM version 2.2.1 and commit 19bf4151d77f97841aacd8d0b743096e48863970. The IGM model set-up from Leger et al., 2025 used in this study is accessible
via https://zenodo.org/records/14275231.



## 7 Author contributions

Conceptualization: HW, DS, support from TPML, GJ. Methodology: HW, DS, TPML, GJ. Investigation: HW, support from DS. Visualization: HW, support from DS and TPML. Supervision: DS, support from RW. Writing (original draft): HW, support from DS. Writing (review and editing): HW, DS, TPML, GJ, RW.

## 8 Competing interests

The authors declare that they have no conflict of interest.

## 9 Acknowledgements

This study was supported by the German Research Foundation (DFG) grant to Dirk Scherler (SCHE 1676/6-1) and Ricarda Winkelmann under the priority program "Mountain Building Processes in Four Dimensions (MB-4D)" and by the European
Research Council under the European Union's Horizon 2020 research and innovation programme under grant agreement 759639. This work utilized high-performance computing resources made possible by funding from the Ministry of Science, Research and Culture of the State of Brandenburg (MWFK) and are operated by the IT Services and Operations unit of the Helmholtz Centre Potsdam.

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
