# Peer review of "Impact of spatial resolution on large-scale ice cover modelling of mountainous regions"

_EGUsphere, 2025_

## Author Comment (AC1)

**Comment on egusphere-2025-3870 (referee #1)**

*We would like to thank reviewer 1 kindly for taking the time to read our manuscript and for the constructive feedback, which will help us to improve the study. In the following, we will address all comments point by point and suggest respective corrections. Please find our answers in blue italic font.*

Werner et al. present a study on the impact of spatial resolution when modelling extensive ice cover of mountainous regions. They find that increasingly coarse resolution leads to greater ice volume, slower ice, and slower response times -- at least part of which can be explained by the coarser resolution lowering peaks and filling in valleys.

I think all of the modelling behind the results and discussion is competently and appropriately done (though one or two points about instructed vs. numerical model implementation) and as such don't believe further simulations are necessary. However, I think there is scope (and need) to really simplify, shorten, and tighten the text and the figures and to focus on providing some key takeaways for practitioners (even if that comes with caveats).

For this reason I think this falls into major revisions, even if no new analysis is needed. I also expect that the current presentation means it is possible that I may not have caught all the issues.

*We thank the reviewer for the general assessment and fully agree that there is a need to shorten and tighten the manuscript. Among other things, repetitions in the results and discussion should be deleted, and the focus should be narrowed down to fewer points. Please see below our responses to other major and minor comments for more details on this point.*

*We also agree that the perspective of modellers should be more expanded. We believe it is unlikely to give very specific recommendations for spatial resolution based on any DEM because input and output variables vary differently to resolution. However, we are confident that by focusing on key takeaways and shifting away from the perspective of "resolution modes", the manuscript will provide more practical use.*

*To significantly shorten the main text, we suggest to remove Figs. 6 (showing velocity difference maps) and 9 (showing resolution effects throughout the cooling phase) and focus on (i) the examination of ice volume changes throughout the run (Figs. 3, 4, 10) as well as (ii) ice thickness, velocity, and basal temperature at the full glaciation (Figs. 5, 7, 8).*

At present the emphasis is on where the highest sensitivity to resolution is (about 400-800 m). This is one way of putting it that could make sense if one has been dealing with interpreting these simulations for a long time. However, if we take 50 m as the 'most realistic' run, then the important value is not so much the rate of change with respect to the previous simulation, but

the total drift away from the most realistic simulation. This information then provides an interested reader with information along the lines of 'yes, you can coarsen your resolution to improve runtime, but you will be dealing then with an error of this rough order'. It feels like the authors are maybe avoiding being more definite about this which is perhaps understandable, but I think you have a good case to provide more concrete recommendations (supplied with suitable caveats). For example, Williams et al. (2025, https://doi.org/10.1038/s43247-025-02010-z) suggest 5 km -- maybe this isn't a perfect number or maybe it's spot on, but at the very least it is a working value that can be used in lieu of further investigation.

I would suggest switching to this approach, meaning figures, results, discussions of the actual variable in question (not a difference), or the error relative to that one baseline value. If the authors feel strongly about keeping their current representation I think some strong reasoning for their approach is required.

*Thank you for the suggestion, we agree that our analysis and most of the figures are suitable for taking the 50 m run as a reference and comparing results to the 50 m reference run; other figures and text passages will be adjusted.*

*Generally, instead of emphasizing the "critical mode of resolution sensitivity" at intermediate resolutions, we suggest to focus on the contrast of model results at fine and coarse resolutions for two main reasons:*

1. *This approach would enable us to consider the 300 m resolution as "sufficiently fine" to obtain model results in close agreement with the 50 m reference run. We believe that this framing is clearer and more useful to modellers. Especially, the resolution of 300 m for the exemplary ice field in the Western Alps could be a starting point and reference for practical resolution considerations.*

2. *Results at fine (50-300 m) and coarse (<800 m) resolutions differ substantially: The ice at fine resolution is more topographically constrained with faster ice flow and thinner glaciers, while it is rather thick and smooth with slower ice flow at coarse resolution. We discuss explanations for these differences from characteristics of the DEM at different bedrock altitudes and resolutions in section 4.1 (which will be revised and shortened), e.g., deeply incised valleys with a more constricted flow at fine resolutions. We believe that the comparison between fine and coarse resolutions and the explanations based on the DEMs are more understandable, visual, and easier to keep in mind than the more abstract concept of the "critical mode".*

Beyond that, I think there is a bit of an oversupply of figures here (and the detail in some, such as 9, is a bit too condensed). I don't want to suggest exactly how this should be done, that requires thought, time, and familiarity with the material that I don't have (nor is it my place) but I would encourage the authors to think about what the key takeaways they want a reader to

have are, and emphasise these in a reduced number of figures (and figure panels). Additional information can go into the supplement, even if that is already rather crowded !

*As mentioned above, we suggest removing Fig. 9 and Fig. 6 which shows maps of ice velocity differences. To follow the approach of taking the 50 m run as a reference, Fig. 5 will be replaced by a reduced version of Fig. S7 which shows ice thickness differences between different resolutions and the 50 m run. Below we show a possible updated version of Fig. 5:*

[Figure]

*We also believe that the analysis of the Arrhenius factor changes with resolutions in Fig. 8f does not add many insights and suggest removing this aspect from Fig. 8 and the discussion.*

I was also struggling to get through the results and part of the discussion. I hope that focussing on a reduced number of central points might make it possible to streamline your findings and paint a clearer narrative. This doesn't mean important details have to be lost, just that the ridges and valleys of the text should be a bit more defined to take a mountainous analogy. As with the figures, I won't go into full details on all of specific ways this could be done in the more minor comments.

*Thank you for raising this point, we agree that the text style should be clearer. In the results, a general removal of figure descriptions and emphasis on interpretations and takeaways will tighten and shorten the text. The reduced number of panels in the suggested design for Fig. 5 (see above) and the removal of panel f in Fig. 8 gives the opportunity to further shorten the corresponding paragraphs. Similarly, corresponding paragraphs to Figs. 6 and 9 can be removed.*

*We also see the potential for a shorter and more focussed discussion, mainly by implementing the following points:*

- *The shift in perspective from focusing on the "critical mode" to comparing results at fine and coarser resolutions is clearer which will be reflected in the text. Consequently, results can be discussed more directly instead of having to describe what the "critical mode" is.*

- *In section 4.1, where we discuss explanations for the resolution-effects across bedrock altitudes, there is repetition of results which will be removed. Additionally, the paragraph on mid altitudes will be combined with the paragraph on low altitudes and focus on the influence of valleys in general.*

- *Generally, the style of the text should be more direct, which will reduce the word count and focus the reader's attention.*

My minor comments are not fully comprehensive as I think there is enough work to be done here that that could be a redundant effort, but I have tried to go into some detail !

To end on a positive note, I think the results are really interesting and that this should eventually produce a very valuable paper. So apologies if the below reads as harsh, it's not my intention. I think you have something good here but it needs a lot of filing down and a change in tack on the presentation front.

**Abstract:**

16 - Opportunity here to say in which direction at no/minor additional word cost.

*Thank you for pointing this out. Our initial idea here was to emphasize the non-linearity of our results before describing the findings at fine and coarse resolutions. By keeping both points separate, we tried to emphasize both of them more clearly. However, we will consider merging those two aspects in the revised manuscript.*

18 - It could be me, but here and elsewhere I had to do a bit of thinking about what the critical mode actually is. Following the major comments this could be redundant, but I would suggest using a more easily graspable phrase here.

*Thank you for this important comment. Following your suggestions above, we think it is best to remove the phrase "critical mode" and emphasize differences between fine and coarse resolutions and viewing the resolutions at 300 m and finer as "sufficiently fine" to accurately capture topography-controlled ice dynamics.*

*We would like to explain our initial idea to define the "critical mode": The "critical mode" referred to results at intermediate resolutions where we found strong changes between the resolution runs, e.g., between simulations at 400 and 500 m resolution. We consider this mode "critical" because modellers that use, e.g., a resolution of 500 m would obtain results quite different to 400 m, although the differences between the DEMs might seem minor and additional computational costs are only slightly lower for 400 m resolution. While it might be obvious that differences between simulations at, e.g., 1 km and 50 m are high, it is less intuitive that results*

*differ significantly between 400 and 500 m resolution. For example, the maximum ice volume from the 400 m run increases by over 25 % when using a 500 m resolution.*

20 - Can say that coarse resolution artificially e.g. lowers slope angles as I'm sure we're all in agreement that coarse resolution is less realistic.

*Thank you for the suggestion, we will adopt that in the revised manuscript, e.g.: "In contrast, topographic resampling to resolutions coarser than ~800 m **artificially** lowers slope angles as well as mountain peaks and raises valley floors, leading to slower-flowing, thicker ice across all altitudes and prolongating glacial response times."*

21 - 'Slower temperature...' I guess this refers to coarse resolution? but should be clearer

*Yes, the reduced hysteresis effects with slower temperature forcing refer to coarser resolutions and the original text passage can be changed to: "Slower temperature change partially reduces the hysteresis between climate forcing and glacial response **at coarse resolutions**…"*

23 - Critical mode could be either defined, or a different term could be used

*We fully agree and will remove the term "critical mode" (see also comments above). We propose to focus on the comparison between fine and coarse resolutions as well as resolutions of 300 m and finer as "sufficiently fine".*

24 - 'We expect similar' could go to 'non-linear... are likely in mountain regions worldwide' as I think you can be reasonably sure of this.

*Good point, we will adopt this: "**Non-linear** and altitudinal-dependent resolution effects **are likely** in mountain regions worldwide and emphasize the need for model advances to enable simulations at sufficiently high spatial resolutions to accurately resolve glacier dynamics."*

Here and elsewhere it is not always super clear in what direction the non-linearity is pointing -- as with the major comments I would suggest honing in on a few of the most significant changes and focussing on clear descriptions of these.

*Thank you for highlighting that previous descriptions of resolution effects were not focussed enough. We would like to mention that in addition to the general trend that coarser resolutions yield more ice volume and slower ice flow, we believe that the resolution effects at different bedrock altitudes are a main takeaway: (i) thicker ice, slower ice flow, and generally frozen bed conditions at mountain tops at fine resolutions, (ii) more constrained and thus faster ice flow in the valleys at fine resolutions compared to slower-flowing and thicker ice at coarse resolutions that is supported by the surface mass balance-elevation feedback. We will make sure to be clearer about the resolution effects and focus on ice volume, thickness, velocity, and basal temperature.*

I might come back to this in the Discussion, but I think the question of whether these changes can be expected in different regions is an interesting one. One thought I had is that an 'idealised'

mountain range could be used for widespread applicability, or you could just run this over the Himalayas too. That does arguably 'fall outside the scope of this study', but even without that I think you can be more confident in your assertion that other mountain ranges should follow similar patterns. Certainly the lower peaks and shallower valleys part is self-evident.

*Thank you for raising this point. In the manuscript, we argue that resolution effects similar to those we found in the Western Alps appear in other (also smoother) regions based on experiments with a smoothed topography (lines 494-507). We believe it is important to point out that the qualitative description of resolution effects in other regions is likely to be very similar to our findings. However, we agree that further research is needed especially to investigate quantitative characteristics, e.g., the magnitude of ice volume differences or find a "sufficiently fine resolution", which is beyond the scope of our study.*

**Introduction:**

I think some broader literature on the importance of resolution is missing from this introduction. In addition to the Williams paper cited above, a quick google scholar search 'importance of resolution for ice sheet modelling' reveals quite a wealth of literature. I think this work should be better situated within this literature, as well as what sets it apart (namely the use of IGGM and more of a focus on mountains). Searching for mountain-glacier specific literature gave less hits, but then you can talk about this being one of the first (if that is the case).

*We thank the reviewer for raising this point and providing the valuable reference. We fully agree and think our work can be motivated by referring to resolution effects for ice-sheet modelling of Greenland and Antarctica (Williams et al., 2025; Aschwanden et al., 2016; Rückamp et al., 2020; Cuzzone et al., 2019), who also connect resolution effects to differences in the DEMs. Because the underlying topography of ice-sheets is relatively smooth while mountainous topography is complex across the whole mountain range, resolution effects in mountain regions are expected to influence model results more. To our knowledge, there exists no resolution recommendation for ice fields and mountain glaciers, which emphasizes the importance of our analysis and further motives changing the narrative from "the critical mode" to 300 m and finer being "sufficiently fine resolutions" to accurately resolve the topography for glacier dynamics.*

33 - Overuse of extensive if you feel inclined to change that.

*Good point, this could be reformulated, e.g., "During the Quaternary period, extensive glaciations carved steep and narrow peaks and formed over-deepened and U-shaped valleys (Ivy-Ochs, 2015; Liebl et al., 2021; Penck and Brückner, 1909)."*

42 - I don't think references are necessary for this statement

*We agree and will remove the references: "For example, elevation controls the mass balance mainly through the temperature lapse-rate, whereas the steepness of the bed controls glacier flow."*

42 - about -> for

*Thank you, we will change this accordingly: "the topographic details **for** peaks and valleys".*

48 - In The Alps? or in our model domain? Region could be specified, basically.

*This sentence refers to the Western Alps: "Specifically, at elevations higher than 1000 m a.s.l. **in the Western Alps**, slope angles in a 100 m DEM are ~5–10 ° steeper compared to a 500 m DEM and even ~10–15 ° steeper compared to a 1000 m DEM (Fig. 1**Fehler! Verweisquelle konnte nicht gefunden werden.**c)."*

65 - Averaged over the entire domain ? or a local maximum ?

*This sentence refers to the Rhone Valley: "In the European Alps, comparisons between modelled ice surface elevations and trimline suggests ice thickness exaggerated during the Last Glacial Maximum (LGM, ~24,000 years ago) in models based on a km-scale resolution, e.g., by 400–1000 m on average **in the Rhône Valley** (Seguinot et al., 2018; Jouvet et al., 2023)."*

66 - 'A better match...' not clear how this sentence supports your argument and also raises the point, is the 400-1000 m exaggeration due to resolution or sliding, or (probably better), you're suggesting that both have a role to play.

*Thank you for raising this point. We propose to add a sentence addressing this comment, e.g.: "A better match between modelled ice surface elevations and trimlines was achieved by Mey et al. (2016), by adjusting the sliding coefficient. Recent work by Leger et al. (2025) significantly reduced the ice thickness offset to ~150 m by using a higher spatial resolution of 300 m in an ensemble of 100 simulations with various parameter combinations. **These advances suggest that the agreement between numerical models and field data can be reduced not only through improved parameter choices but also by increasing the spatial resolution.** However, it remains unclear whether even finer resolutions lead to more accurate results, and how spatial resolution influences other aspects of glacier modelling in alpine regions."*

77 - This passage maybe trails off a bit -- I don't think you have to mention everything you do, but at the same time I generally find the use of 'other key variables' a bit vague. If there aren't so many 'other variables', then it's maybe best to simply list all of them, or following the major comments to be more specific about which you take as key indicators (and why).

*We agree and following the comments above, it should be clarified that we consider the temporal evolution of ice volume, as well as spatial and average patterns in thickness, velocity, and basal temperature at full glaciation for our analysis.*

**Methods:**

Good to talk about the implications of using IGGM over a full numerical model. How big could the innacuracies be? Your approach is certainly justified, but as IGGM is quite new, some more background (this part in the intro) would be good!

*Thank you for this comment. We will address this point below and believe that more information on specific aspects of IGM, especially the retraining frequency will help to understand IGM and its limitations better.*

*In general, an extensive description with examples for applications, and a comparison between IGM and analytical solutions is provided by Jouvet and Cordonnier (2023). IGM makes uses a neural network emulator to model ice flow. Similarly to traditional model, this neural network is trained to minimize the energy associated with Blatter-Pattyn model. As glaciers evolve in time, the ice flow emulator needs to be frequently retrained to adapt to the new conditions. Therefore, the retraining frequency plays an important role to mitigate inaccuracies (i.e., the degree of assimilation of the Blatter-Pattyn model). We aim to find a trade-off between accuracy (that require enough retraining) and computational cost (that require to mitigate the retraining). We will make sure that the essential functionality of IGM is clear from the text and we will address inaccuracies based on our model choices in the revised manuscript.*

88 - Can you give a more quantitive description of its accuracy ?

*As pointed out above, inaccuracies of IGM are discussed in Jouvet and Cordonnier (2023) and the retraining frequency is the most important parameter to account for inaccuracies. The retraining frequencies we use for our simulations correspond to retraining every 0.02-0.28 years, meaning that the emulator is retrained based on the Blatter-Pattyn model multiple times a year, reducing possible inaccuracies.*

89 - End of this sentence a bit unclears

*Thank you for this comment, we propose to replace the end of the sentence with the following to enhance clarity: "The physics-informed deep learning approach is independent of training data from other models, which allows us to run the same experimental set-up at different spatial resolutions and directly compare the model results."*

90 - A few more details here about Leger et al. would be good. Particularly the comparisons to actual physics based models, and to observational data. This is the core of your results, so it's good for the reader to know the underlying model can be trusted (even if it is an unseeing unthinking statistical emulation :p).

*We would like to mention that a comparison between results from Leger et al. (2025) and observational data can be found in the introduction: "Recent work by Leger et al. (2025) significantly reduced the ice thickness offset to ~150 m by using a higher spatial resolution of*

*300 m in an ensemble with 100 parameter-perturbed members." Additionally we find it valuable to add to the manuscript that Leger et al., 2025 validated an IGM simulation of the European Alps at 2 km resolution by comparing it directly to the simulation (also at 2 km) using the Parallel Ice Sheet Model (PISM; Winkelmann et al., 2011) published by Jouvet et al. (2023), which should strengthen the reader's trust in our model setup.*

99 - 'high-order' -> higher-order

*Yes, we will change the sentence accordingly: "The ice velocities are simulated as energy-minimizing solutions of the **higher-order** Blatter-Pattyn model (Blatter, 1995)…".*

100 - hopefully not a killer question, but is the performance a function of the model resolution used in training ? I guess the training takes quite a long time to run so this is a bit difficult to test, but could be quite important. For example, if training at a much lower resolution mitigates many of the problems associated with low actual resolution, that's a pretty important result. Some discussion/thoughts on this would be good !

*Thank you for this interesting question. In general, the idea behind the pretraining in IGM is to make sure that the simulated ice flow converges with the Blatter-Patty solution more quickly. With enough retraining, the ice flow should not be dependent on the pretraining after the beginning of the simulation. So, the pretraining only affects only the first time steps of the simulation.*

*Since the pretraining is based on a fine resolution of 100 m and the retraining frequency during run times is quite high, the resolution in the pretraining is not expected to affect model results. Moreover, possible adjustments in the 50 m run to the glacial state from pretraining at 100 m are mitigated during the 1000-year initialization phase. Similarly, deviations from a pretrained emulator at coarse resolution would be accounted for during initialization. We believe an extensive explanation of the implications of spatial resolutions during pretraining is not necessary, but a mention that those possible deviations are accounted for in the initialization will add clarity.*

104 - It's worth making it clear that Jouvet and Cordonnier did this and not you (if that is the case). See also the comment about introducing Leger et al. in a bit more detail (l. 90)

*We agree and propose to add the references to both sentences: "The initial glacial state is based on a pretrained CNN described in Jouvet and Cordonnier (2023) to help with convergence to the Blatter-Patty model. The emulator was trained over a diverse catalogue of glaciers and flow regimes at 100 m spatial resolution with 10 vertical layers, fine vertical discretization close to the ice-bedrock interface, 16 CNN-layers, and 32 CNN-output filters **(Jouvet and Cordonnier, 2023)**."*

106 - How was it retrained?

*The retraining takes place during run times and is physics-informed, i.e., the weights the Convolutional Neural Network (CNN), which predicts ice flow velocities, are optimized to minimize the energy associated with the Blatter-Pattyn model (Jouvet and Cordonnier, 2023). As mentioned above, we believe that clarifying the functionality is needed to understand the essentials of IGM without giving a detailed description of the model.*

130 - I guess using a spatially inextensive ice sheet ? Shouldn't really matter as you're testing sensitivity only (not gaining 'actual' results) but could be worth mentioning.

*We are not sure that we understand where this question is pointing at. Our modelled ice field area is ~12,700 km$^2$, which is comparable in size to the present-day Southern Patagonian Icefield (lines 146-147).*

*The gflex module described in lines 126-130 is applied over the whole model domain. In order to lower the computational costs the spatial resolution of the gflex module is lowered to 2 km, which is however fine enough to capture large-scale GIA-induced changes, as shown by Leger et al. (2025).*

151 - Not sure I catch the drift of this sentence

*Thank you for pointing out that more explanations are needed. This sentence refers to experiments with a smoothed topography are not shown in the results but are used only as reference in the discussion. Therefore, we propose to delete this sentence from the main text and explain those experiments in the supplements (where corresponding figures are shown) in more detail: "The following two figure show input and outputs from additional experiments based on a smoothed topography. The smoothed topography is obtained through the following process: We took the DEM of our model domain at 2 km resolution, which was created from the fine resolution DEM using cubic resampling. Therefore, the 2 km DEM is rather smooth with less topographic detail. We then proceeded to interpolate the 2 km DEM back to finer resolutions, again using cubic convolution."*

156 - You could specify you start 'warm', or if tied to the present day temperature, how cold exactly. Also good to have a very brief idea of what ice configuration this sets the model up in. Your time step is 0.01-0.04 -- when does this value change? Why do you use this value ?

*Thank you for the suggestion. Following the next comment, the present-day climate can be specified by giving an annual temperature that remains unchanged during the initialization phase and is the starting point for the cooling phase. As written in line 152, the initial ice configuration is that every simulation starts without any ice.*

*We will specify the ice volume values at the time after the initialization when the simulations are in balance with present-day climate in section 3.2: "After the initialization phase (0–1000 years), the **2000 m simulation has ~160 km$^2$ of ice volume** which is more than 10 times that of the **50 m run (~12 km$^2$)."***

*The time step in IGM is adaptive and not static. IGM computes the time step such that it satisfies the CFL convergence condition to ensure numerical stability for the ice thickness evolution. Therefore, the time step is not a chosen model parameter and instead calculated internally. We propose to remove the details on the time stepping in IGM from the revised manuscript and use the unit years for the retraining frequencies instead: "For adjustments to new ice field states attained through run time, we retrained the physics-informed neural network multiple times within a single model year (every ~0.18 years). The on-the-run retraining in the highest-resolution simulations at 50 and 100 m was even performed more often (every ~0.05 years)."*

165 - This lapse rate opens the possibility of reporting an annual temperature at sea level, which is a bit more human redable than degree Kelvin away from the base temperature.

*Yes, we would like to add that the mean annual temperature at sea level is ~14.9 °C, which we refer to as "present-day climate" (i.e., no temperature change applied).*

165 - I don't follow this statement exactly. It was multiplied by 1.6 across all times/temperatures, or just increased from 1 to 1.6 as things got colder? I fully accept the main aim of this paper is to test out resolution, but given you are emulating a realistic event, it would be nice to know a bit more information/rational behind this step.

*Thank you, we agree that more clarification is needed. We propose to change the text passage to: "The weather station's annual precipitation of 660 mm lies at the lower end of typical values in the European Alps (e.g., Isotta et al., 2014).  To ensure that the ice field covers the entire mountainous part of the model domain at the time of maximum cooling, we therefore multiplied monthly precipitation values by 1.6, resulting in an annual precipitation of 1056 mm which is more in line with the average range. Those monthly precipitation values were kept constant in space and unchanged throughout the warming and cooling phases."*

167 - This statement is implicit if you don't feel like including it

*Thank you for the comment, we believe this statement might be helpful for the reader to make sure they do not confuse our experiments with an attempt for realistic simulations of the LGM.*

176 - What percentage of your domain did this influence ? Was this applied for all resolutions if only one resolution qualified ?

*Thank you for raising these questions. The mask is based on the 50 m resolution run and it is applied to all simulations after resampling all model outputs to 50 m (see lines 171-172). Please note that due to the choice of the model region, the mask is only effective at the left and right boundaries because the ice does not reach top and bottom boundaries at all resolution. At 50 m resolution, ice-covered pixels (at full glaciation) outside of the mask are ~14 % of the entire model domain. Because the ice field is slightly more extensive towards the top and bottom of the model domain at coarser resolution, the influence of the mask is expected to be lower at these resolutions.*

*We decided on this approach to make the comparisons between the resolutions more straightforward and not depend on a watershed algorithm that determines the ice area for each resolution individually. Moreover, because the 50 m run serves as our reference, we like to base the model setup (in this case the area considered for our analysis) for all resolutions on the 50 m run.*

**Results:**

Please also see my major comment on this. I think there is significant scope to streamline this, and I don't think it's my place to point out every opportunity so I leave that to you, hence the not huge number of individual comments.

185 - Assumption to -> assessment? I get that this is a bit of a grey area but then again, there is enough of a background that you probably could call this an assessment not an assumption.

*Thank you for this comment. To avoid assessment and assess in the same sentence, we propose to use "notion": "Based on the **notion** that resampling to a coarser resolution affects large and deep valleys less compared to small tributary gorges or ravines, we assessed elevation differences for valleys of distinct Strahler order (Strahler, 1957)."*

190 - With reference to the 30 m base map ?

*The reference for all input and output comparisons is the 50 m resolution. Since this might not be clear from the text, we suggest to add that information: "The highest discrepancies are at intermediate Strahler orders (3–4), with median difference values ranging from 150 to over 250 m for DEMs resampled to a resolution of 1000 m and coarser **compared to the 50 m DEM**."*

193 - A little confused by this sentence ! I think it could do with some reorganisation/clarification.

*We agree and propose change this sentence to: "In contrast, at very low (1) or high (6–7) Strahler orders, median difference values are less than 165 m, for any resolution."*

Fig. 2 - I also think you might need a supplementary figure showing an example of how these stream regions are classified -- from watershed to watershed ?

*Thank you for the suggestion, we will consider adding such a supplementary figure.*

194 - I am no master statistician, but if all the changes are positive, but the box and whisker plot goes negative, then perhaps a different range indication is called for?

*The sentence ("Notably, elevation differences are strictly positive at these stream orders, indicating resampling-induced elevated valley floors in larger valleys.") refers to high stream orders (6-7), where Figure 2 shows only positive values, especially for the box and whiskers.*

205 - In runs at all resolutions ?

*Thank you for the suggestion, this expression appears a few times in the manuscript. We will check the whole manuscript and change it to "runs at all resolutions" or "simulations at all resolutions" in these cases.*

210 - Can this be a range?

*Thank you for the comment, this question applies to other passages in the text as well (see also the next comment). In line with our suggestion to handle one of your major comments on the "critical mode", we would like to emphasize the differences between fine and coarse resolution simulations. To clarify the resolution ranges for "fine" and "coarse" we think it is best to use consistent resolution ranges, where ice field dynamics are similar, e.g., 50-300 m (fine resolution) and 900-2000 m (coarse resolution). For this specific text passage, we propose "simulations of 300 m and finer".*

215 - This whole paragraph could do with tightening up a bit (sorry I know it's always hard to strike a balance between detail and brevity !). I think the reason it's a little hard to follow is that the values in question are jumping around a bit and the distinctions can feel a bit arbitrary. I.e. sometimes it's 1000 m and the 300 and 400 m, and then it's <500 and >800 . And then at line 221 it goes to 600 and 800. Good to be consistent. May also suggest just three exact values, rather than ranges, which can be a bit harder to describe. I thought about moving this to a major comment but it should be easy to rectify. I do think consistent resolution ranges across the paper are important/useful though !

*In line with the above comment, we agree to use consistent resolution thresholds. Additionally, we see the potential to tighten this paragraph with more direct and less descriptive sentences. We believe that the most important point of this paragraph, being that the finer resolutions capture changes in temperature forcing more rapidly, should be clearer.*

217 - The sentence beginning on this line is quite unclear

*We agree and as this sentence does not add much new insights, it can be removed.*

221 - I think I got what's going on here, but it required more interpretation than I would like. the '-' between warming and cooling is a minus sign? At first glance it looks like punctuation. So either an explicit 'minus' or using more obviously mathematical variables for warming and cooling would be good.

*Thank you for the suggestion, we will modify the figure accordingly (see end of this document).*

223 - % and deg C brake convection by coming directly after the number

*Thank you for the comment, we will check all units and align them with the conventions.*

225 - So in effect, the higher resolution simulations are much more sensitive to climate change -- that could be a headline finding !

*In line with the comment on line 215, we agree that this should be clearer from the results. Additionally, we see the potential to (slightly) expand on this in the discussion, section 4.2.*

255 - The first two sentences of this paragraph feel a little winding, maybe there is a punchier way to arrive at your descriptions ? I will confess I did not manage to read all of the paragraphs around this point word for word because they are very dense.

*We agree. This comment refers to the paragraph on Fig. 5 which will be adjusted such that it shows thickness differences with reference to the 50 m run (see proposed figure above). When adjusting this whole paragraph, we will make sure to tighten it (especially the beginning).*

264 - I guess this comes back to my earlier comments, it would be nice to have more consistency in resolutions comparisons, that might make a headline figure /narrative of 'use this resolution if you would like to avoid the worst resolution issues'. I do accept that reality will be a little bit more complicated, but here a little more synthesis is not unwarranted

*Similarly to the previous comment, there are a lot of changes expected to happen in this paragraph along with updated figures. Nevertheless, we believe that the spatial distribution of ice thickness differences clearly shows that differences are small among the fine resolutions (50-300 m), indicating that 300 m is a sufficiently fine resolution for our exemplary ice field. We will make sure to emphasize this finding in the revised manuscript.*

267 - And below? So, it only becomes a clear piedmont glacier as resolution decreases from 900 - 500 m

*Thank you for raising this question. At fine resolutions, the length of this specific glacier is too short to form a piedmont-type terminus (Fig. S3). However, we consider the observations of piedmont-type glaciers a minor detail and suggest to remove this part from the manuscript to focus on main aspects.*

282 - I would argue this should be reinterpreted. We can say that the lowest resolution is the most accurate, so then the importance is the drift away from that, the difference between the resolutions is then perhaps of more supplementary interest. I would lessen the number of panels (even if having a panel o is cool), but also include one that shows the actual absolute field at 50 m resolution for comparison purposes.

*As mentioned above, we propose to remove this paragraph and figure from the main text. For a comparison of 2D ice velocities to the 50 m reference run, we refer to the supplements (Fig. S6).*

288 - when compared. This sentence is very long-winded.

*We agree that the beginning of this paragraph should be more concise. Moreover, because we will compare only to the 50 m reference run and not between resolution, this sentence will not be needed and will be removed.*

290-390 - I should be honest and say that I am having a very difficult time reading through all this and I am giving it a good go. I have managed a skim but I think my main recommendation would be a focussed rewrite. For example, you switch to ice temperature at 327 after a lot of text about slope values and DEMs. That certainly should be its own paragraph, or grouped in a shorter paragraph with something clearly closely related. All this to say, I expect this can be really significantly cut down to focus on the main takeaways, and a similar approach applied to the rest of the paper. If you put in a few easy to decipher figures they can do the talking, and the results text can be much reduced.

*Thank you for this comment. As pointed out previously, we agree and think it is best to remove Fig. 6, 9, panel 8f and the corresponding paragraphs. Additionally, we generally see the potential to tighten the text while presenting main takeaways more directly. In this specific paragraph, we propose to make the connection between the input variables (elevation and slope) and the modelled variables clearer, which should focus the reader's attention better.*

**Discussion:**

As with the rest of the paper, this could be very much streamlined, with a big focus on what the main takeaways are. I think this could go from 8 to at most 5 paragraphs, hopefully less. Implications, and tying this in to the rest of the literature should really be emphasised. There are few citations in this section.

*Thank you for this comment. We also see the potential to reduce the number of paragraphs and shortened some paragraphs by removing repetitions from the results. For example, the paragraphs in section 4.1 on mid and low altitudes can be combined and shorten. Additionally, a brief comparison of our results to findings from modelling Greenland and Antarctica will be inserted to put our study in a broader context. Moreover, the effect of faster ice flow at fine resolutions is also discussed by Leger et al. (2025) and Cuzzone et al. (2019) and will be cited.*

405: The last sentence of this paragraph is useful (could actually be expanded so that reference back to Fig. 5 is not necessary). I think the rest can be restructured around distance from 50 m and the points you really want to emphasise .

*As mentioned above, the comparisons of resolutions will be analyzed with respect to the 50 m run. The spatial pattern of resolution differences between mountain tops and valleys is discussed in section 4.1 (which will be tighter, shorter, and more focused).*

418 - Here and elsewhere you can drop 'apparent'. You're controlling every variable in this model so you can say this with a modicum of conviction.

*Good point, we will check for this term and drop it where relevant.*

421 - Ditto, seem.

*We agree and will check the whole paragraph.*

423 - Basal melting will only happen if the temperature is at the pressure melting point, so this statement is a bit vague.

*Thank you for this comment. The higher probability for basal melting is just one possible explanation for the connection between higher ice thickness and warmer temperatures at coarse resolutions (lines 423-426). Our results in Fig. 8 show that mean basal temperatures at coarse resolution are at 0 °C and thus closer to the pressure melting point than at finer resolution. However, since these are mean temperatures, we argue it is justified to be a bit vague about this. For clarifications we suggest to rewrite this part, e.g.: "The shift from frozen bed conditions at fine resolution simulations to temperate basal ice at coarser resolutions reflects colder climatic conditions in the fine resolution DEMs due to higher surface elevations. **Moreover, the thermal shift is supported by increased thickness at coarser resolutions with a higher likelihood** for basal melting due to increased basal pressure and thus a lower pressure melting point."*

424 - Taking an uncomplicated 1D column shear is a \rho g h sin(\alpha), so it's a function of height and slope. Greater thickness doesn't necessarily mean equal or greater slope.

*Thank you for the clarification. It is true that we cannot conclude that internal deformation is higher at coarser resolution due to thicker ice, especially because mean slopes are indeed lower (Fig. S10). We will remove the corresponding text passage from the revised manuscript.*

425 - Could be clearer.

*We agree and propose to change the sentence to: "The shift from frozen bed conditions at fine resolution simulations to temperate basal ice at coarser resolutions **reflects colder climatic conditions in the fine resolution DEMs due to higher surface elevations**."*

426 - Better in results ? Link of temperature to Arrhenius is well known, so you could just focus on temperature.

*We agree and will remove the parts about the Arrhenius factor to enhance the focus of the paper.*

428 - This definitely does not need a presumably (softness dependence on temperature)

*We agree and we think it to be best to remove the analysis of softness (see comment above).*

432 - 437 - Results ? But please think carefully about adding to the results !

*We think this text passage can be removed as it is already part of the results.*

437 - Repetition ?

*Good point, we will remove this from the discussion.*

445-447 - Results ?

*This is already part of the results and will be removed.*

447 - 449 - Cool

*Thank you for this comment.*

452 - 455 Results ? or Not that relevant

*We agree to move the specific number for velocity differences to the results. Additionally, this paragraph will be shorter and the discussion on the velocity among fine resolutions will be reduced from three sentences to: "In contrast, the decrease of ice velocity among the finest resolutions might result from increased lateral shear stress from very steep and narrow valley walls, imposing resistance and slowing down ice flow."*

459 - Again with seems, hopefully you can be a bit more definite about this. Interesting discussion tough

*We agree and this sentence will be changed to: "This feedback is most likely strongest in the valleys with steep and narrow sidewalls at fine resolution and partly offsets ice thinning due to acceleration from channelized flow, indicated by consistently thicker ice at low altitudes than at high and mid altitudes - across all resolutions (Fig. 7)."*

471 - Half as fast, maybe twice as slow, but not half faster or twice slower, somehow. Probably 'when the rate of temperature forcing is reduced by half'

*Thank you for this suggestion, we will change the sentence accordingly.*

4.2 Opening sentence is results

*We agree, the opening sentence will be removed and the paragraph started by: "Our simulations show that changes in temperature forcing are more accurately captured in the fine resolution simulations which respond more rapidly to transitions from cooling to warming periods (Figs. 3c, 10)."*

I'm going to stop going through this line by line here. Hopefully some of the issues are clear from where I have gone line by line above! There is too much mixing of results and discussion. Each paragraph should have a really clear aim and much more direct. For example the hysterisis effect is cool, but then the main thing to emphasise is that hysterisis is reduced for higher resolutions. That, and its implications should be the main focus of this paragraph, not describing that 'up to 420 years compared to a time lag of less than 200 years at the finest resolutions'

*Thank you for the provided line-by-line comments, we will check the rest of the discussion and remove the parts that are describing results to focus more directly on the discussion.*

**Conclusions:**

A conclusion table could be useful. A lot of the bullet points are different ways of saying 'at high resolution this' 'at low resolution this' 'difference is this'. Standard prose styles dictate that we don't repeat language, which I suppose is reasonable, but that really would be useful here because otherwise one has to search through the sentences for the relevant piece of information.

*Thank you for the suggestion. We tried to make a compact conclusion table but ended up repeating results for fine and coarse resolutions in respective columns (with one column oppositional to the other). We will keep this idea in mind and otherwise tighten the bullet points to reduce repetition.*

**Figures**

Please see major comments, but I do feel that the total number of figures can be streamlined into a smaller number of punchier ones. For me the ones which are very useful are 7 and 3 (and maybe something like 6 but for distance from 50 m, and without requiring quite so many panels), but that's just a quick thought !.

Fig. 1 Panel a could be made a bit more recognisable to people through the use of a terrain colourmap and including a couple of city locations -- just a suggestion though.

*Thanks for the suggestion, however, we do believe that Figure 1a is already quite crowded. Moreover, the standard terrain colormap (see below) looks less aesthetic for our taste, and is less colour-blind friendly, that is why we use a custom colormap.*

[Figure]

Fig. 2 - It would be nice to see an example (if not the entire region) with labelled stream order so that one can get a feel for which stream order refers to which type of feature.

*Thank you for the suggestion. We will provide such a figure.*

Fig. 3 - I accept this is a bit pedantic on my part, but could the colour bar be flipped to be in ascending order? Also, 2000 is hard to see so minor suggestion to clip at 1,800 colour (don't worry if that's too much hassle)

Panel c, can't this just be described as the difference as the maximum ice volume will cancel if both are relative to that.

*Thank you for the suggestion, here is an updated version of Fig. 3:*

[Figure]

*We agree that the graph would look the same if we used [ice volume at warming phase "minus" ice volume at cooling phase] compared to that same difference as a percentage of total ice volume. However, we argue that the values of xx % more ice volume in the warming than in the cooling phase relative to total ice volume might be more meaningful than xx km² more ice volume in the warming than in the cooling phase. The reader would have to compare with figure 3b to see if xx km² difference in ice volume is actually a lot or not.*

Fig. 5 This is a valuable figure, but I think it probably could/should be transported to the supplementary material -- a reduced version showing the general trend or important jumps with a bit less white space could reside here

*We agree. A reduced version that compares ice thickness differences compared to the 50 m reference run (Fig. S7) can be found at the beginning of this document.*

Fig. 6 - The pedant within me would like to see Fig. 5 and 6 following the same layout, and I prefer the more compact Fig. 6 layout.

*As mentioned in the comment above, we will use a more compact layout for Fig. 5. We believe that Fig. 6 should be removed from the main text to reduce the number of figures.*

Fig. 7 - I don't exactly understand the positive negative distinction here. All the positive and negative cells binned together ? I would consider omitting that.

*The idea behind the distinction between positive and negative differences is to not mix red (positive) and blue (negative) pixels from Figure 5. To show that patterns at bedrock altitudes are not obscured by averaging over positive and negative values, we argue to take the averages separately.*

Fig. 8 - Can you not develop some improved notation for r? Such as r∈{50,100:100:1000,1200:200:2000} defined early on, and then saying for each r. Writing out the full range at every mention is a bit much. Why is there a grey horizontal line at the top ? I think to separate model inputs from outputs, but then good to label that in the figure. Otherwise a figure like this really does tell story well (though each title should be 'mean ice thickness' etc. ). A panel for total ice volume would be good, too.

*Thank you for the suggestion. We agree that listing all resolutions is unnecessary long. Since the individual values for each resolution are visible from the figure (with resolution on the x-axis and values marked as circles) and resolutions used for our simulations are listed in the methods, we do not think it necessary to mention which resolutions are shown.*

*As mentioned above, we think that the analysis of the mean Arrhenius values does not add much value and propose the following updated figure:*

[Figure]

Fig. 9 The Hr - H50 approach is good. There should be a description on the colour bar. The images are a bit too small.

*Thank you for the suggestions. As mentioned above, we believe that this figure should be removed from the main text.*

Fig. 10, I'm having difficult with this as it feels like temperature is just a proxy for time here? A data axis should not decrease and then increase again. I would suggest separating this out into

two rows and then using two x axis, one for temperature, and another for time. The trend is broadly consistent, so you could also drop the high, mid, low separations.

*Thank you for this comment. Since we applied a linear temperature forcing, model years and temperature values are very closely related. Because in this figure we show results for experiments with different temperature rates, the cooling and warming phases are not equally long. For example, with fast temperature forcing, the cooling phase ends at 3000 years and with slow temperature forcing, it ends at 5000 years. Therefore, we propose to put relative time on the x-axis, from 0 (start of cooling phase with a temperature forcing of 0 K) to 1 (end of cooling phase with a temperature forcing of 0 K). At 0.5, half of the simulation is reached, where temperature forcing peaks at 8 K and the warming phases begin.*

[Figure]

*We argue to keep the distinction between altitudes, which shows interesting details such as different rates of ice volume change with time as well as differences in the timing of maximum ice volume at bedrock altitudes.*

**Supplement:**

Given the quite major comments with the rest of the manuscript I have not extensively explored the supplement. It's nice to see all the parameter values in there. You could put some figures from the main text in here, though even the supplement shouldn't be too long and wavering.

*Thank you for this comment. We will check the supplementary and possibly remove figures from it.*

References

Aschwanden, A., Fahnestock, M. A., and Truffer, M.: Complex Greenland outlet glacier flow captured, Nat Commun, 7, 10524, https://doi.org/10.1038/ncomms10524, 2016.

Cuzzone, J. K., Schlegel, N.-J., Morlighem, M., Larour, E., Briner, J. P., Seroussi, H., and Caron, L.: The impact of model resolution on the simulated Holocene retreat of the southwestern Greenland ice sheet using the Ice Sheet System Model (ISSM), The Cryosphere, 13, 879–893, https://doi.org/10.5194/tc-13-879-2019, 2019.

Egholm, D. L., Knudsen, M. F., Clark, C. D., and Lesemann, J. E.: Modeling the flow of glaciers in steep terrains: The integrated second-order shallow ice approximation (iSOSIA), J. Geophys. Res., 116, 2010JF001900, https://doi.org/10.1029/2010JF001900, 2011.

Imhof, M. A., Cohen, D., Seguinot, J., Aschwanden, A., Funk, M., and Jouvet, G.: Modelling a paleo valley glacier network using a hybrid model: an assessment with a Stokes ice flow model, J. Glaciol., 65, 1000–1010, https://doi.org/10.1017/jog.2019.77, 2019.

Ivy-Ochs, S.: Glacier variations in the European Alps at the end of the last glaciation, CIG, 41, 295–315, https://doi.org/10.18172/cig.2750, 2015.

Jouvet, G. and Cordonnier, G.: Ice-flow model emulator based on physics-informed deep learning, J. Glaciol., 69, 1941–1955, https://doi.org/10.1017/jog.2023.73, 2023.

Jouvet, G., Cohen, D., Russo, E., Buzan, J., Raible, C. C., Haeberli, W., Kamleitner, S., Ivy-Ochs, S., Imhof, M. A., Becker, J. K., Landgraf, A., and Fischer, U. H.: Coupled climate-glacier modelling of the last glaciation in the Alps, J. Glaciol., 69, 1956–1970, https://doi.org/10.1017/jog.2023.74, 2023.

Leger, T. P. M., Jouvet, G., Kamleitner, S., Mey, J., Herman, F., Finley, B. D., Ivy-Ochs, S., Vieli, A., Henz, A., and Nussbaumer, S. U.: A data-consistent model of the last glaciation in the Alps achieved with physics-driven AI, Nat Commun, 16, 848, https://doi.org/10.1038/s41467-025-56168-3, 2025.

Liebl, M., Robl, J., Egholm, D. L., Prasicek, G., Stüwe, K., Gradwohl, G., and Hergarten, S.: Topographic signatures of progressive glacial landscape transformation, Earth Surf Processes Landf, 46, 1964–1980, https://doi.org/10.1002/esp.5139, 2021.

Mey, J., Scherler, D., Wickert, A. D., Egholm, D. L., Tesauro, M., Schildgen, T. F., and Strecker, M. R.: Glacial isostatic uplift of the European Alps, Nat Commun, 7, 13382, https://doi.org/10.1038/ncomms13382, 2016.

Penck, A. and Brückner, E.: Die Alpen im Eiszeitalter, Tauchnitz, 540 pp., 1909.

Rückamp, M., Goelzer, H., and Humbert, A.: Sensitivity of Greenland ice sheet projections to spatial resolution in higher-order simulations: the Alfred Wegener Institute (AWI) contribution to

ISMIP6 Greenland using the Ice-sheet and Sea-level System Model (ISSM), The Cryosphere, 14, 3309–3327, https://doi.org/10.5194/tc-14-3309-2020, 2020.

Rückamp, M., Kleiner, T., and Humbert, A.: Comparison of ice dynamics using full-Stokes and Blatter–Pattyn approximation: application to the Northeast Greenland Ice Stream, The Cryosphere, 16, 1675–1696, https://doi.org/10.5194/tc-16-1675-2022, 2022.

Williams, C. R., Thodoroff, P., Arthern, R. J., Byrne, J., Hosking, J. S., Kaiser, M., Lawrence, N. D., and Kazlauskaite, I.: Calculations of extreme sea level rise scenarios are strongly dependent on ice sheet model resolution, Commun Earth Environ, 6, 60, https://doi.org/10.1038/s43247-025-02010-z, 2025.

Winkelmann, R., Martin, M. A., Haseloff, M., Albrecht, T., Bueler, E., Khroulev, C., and Levermann, A.: The Potsdam Parallel Ice Sheet Model (PISM-PIK) – Part 1: Model description, The Cryosphere, 5, 715–726, https://doi.org/10.5194/tc-5-715-2011, 2011.

Yan, Z., Leng, W., Wang, Y., Xiao, C., and Zhang, T.: A comparison between three-dimensional, transient, thermomechanically coupled first-order and Stokes ice flow models, Journal of Glaciology, 69, 513–524, https://doi.org/10.1017/jog.2022.77, 2023.

---

## Author Comment (AC2)

**Comment on egusphere-2025-3870 (referee #2)**

*We would like to thank reviewer 2 kindly for taking the time to read our manuscript and for the constructive feedback, which will help us to improve the study. In the following, we will address all comments point by point and suggest respective corrections. Please find our answers in blue italic font.*

**General Assessment**

Overall, the manuscript is well-written and the sections are clearly defined. The topic is relevant and timely, given the increasing computational capacity that now allows high-resolution simulations of mountain glaciers. The approach is interesting, and the study addresses an important question: how spatial resolution affects the simulation of ice thickness, flow, and thermal regimes in mountainous regions.

However, while the manuscript is organized into clear sections, some aspects of the results and discussion are difficult to follow, partly due to reliance on references to supplementary material. Certain patterns and the interpretation of figures could be presented more cohesively in the main text.

Besides, there are aspects that require clarification or more explicit justification. While the authors explore a wide range of resolutions (50 m to 2 km), the study relies on a single model (the Instructed Glacier Model, IGM). It is not clear whether generalizations about the impact of resolution on mountain glaciers are fully warranted based on this single framework. Explicit discussion of model limitations and how the resolution interacts with the underlying Blatter-Pattyn approximation, compared to other approaches (full Stokes or others), would strengthen the study.

*We thank the reviewer for the general assessment. We fully agree that the manuscript should be clearer, shorter and not all references to the supplementary material are needed. Additionally, we believe that slightly more background information on IGM, its functionality and limitations will help to put our results into perspective.*

*We would like to clarify that the IGM framework builds on the Blatter-Pattyn model which cannot be replaced by other ice physics approximations or a full Stokes implementation. Due to substantial higher costs of higher-order models, it is computationally too expensive to run our experiments with a numerical, standard and CPU-based Blatter-Pattyn or Full Stokes model (e.g., Elmer Ice). Therefore, it would be impossible to compare our experiments to corresponding simulations using a full Stokes model. To justify using IGM that depends on the Blatter-Pattyn model rather than a full Stokes model, we refer to Pattyn et al. (2008) who showed that the Blatter-Pattyn model generally reproduces ice velocities that agree well with full Stokes models.*

*We believe that a comparison between the Blatter-Patty model and other approaches goes beyond the scope of the study, which is focused on the effect of the spatial resolution rather than the simplification of Blatter-Pattyn vs. Stokes. Instead, we believe that a more extensive comparison to other studies on resolution effects in Greenland and Antarctica (Williams et al., 2025; Rückamp et al., 2022; Aschwanden et al., 2016; Cuzzone et al., 2019) will help to generalize our findings (see other comments below and the review by referee 1 for further details).*

*For a detailed description of IGM and comparisons to an analytical solver, we would like to refer to Jouvet and Cordonnier (2023) and point out that the retraining frequency is the most critical parameter that ensures that the ice flow does not deviate too much from the Blatter-Pattyn model. In our experiments, the emulator is retrained every 0.02-0.28 (model) years, i.e., every 1-2 weeks. Moreover, our setup is based on Leger et al. (2025) who compared an IGM simulation of the European Alps at 2 km resolution to the respective simulation using the Parallel Ice Sheet Model (PISM; Winkelmann et al., 2011) published by Jouvet et al. (2023) and found minor deviations. We believe that adding this information will help to address inaccuracies of IGM.*

**Introduction**

- The introduction's opening paragraph reads as a series of descriptive statements rather than a connected narrative. Consider revising for smoother flow.

*We agree, the importance of ice models to model present-day climate change as well as paleo-reconstructions will be presented with a smoother flow in that paragraph.*

- The knowledge gap and justification for the study could be clarified more explicitly. For example, are there other studies on the effect of the spatial resolution using similar or different approaches or applied to a different setting?

*Thank you for this raising this important point. The impact of resolution has been studied for the Greenland and Antarctic ice-sheets (Williams et al., 2025; Rückamp et al., 2020; Cuzzone et al., 2019; Aschwanden et al., 2016). The analysis by Williams et al. (2025) is based on the Wavelet-based Adaptive-grid Vertically-integrated Ice-sheet model (WAVI), therefore using ice velocity not in three dimensions (like IGM). Results from Rückamp et al,( 2020) are based on ISMIP6 (Ice-Sheet Model Intercomparison Project) experiments, i.e., using models of various complexities. Cuzzone et al. (2019) uses the Ice Sheet System Model (ISSM) at different resolutions based on the Blatter Pattyn model. Aschwanden et al. (2016) compares simulations of Greenland outlet glaciers using the Parallel Ice Sheet model (PISM) that combines the Shallow Ice Approximation (SIA) and the Shallow Shelf Approximation (SSA) at different resolutions. We believe that despite different approaches and although ice dynamics are different for ice-sheets than for mountain glaciers and ice fields, these studies are useful to put our analysis into context. To our knowledge, there exists no systematic resolution analysis for ice fields and mountain glaciers, which emphasizes the importance of our study.*

**Methodology**

- The choice of model parameters requires more justification. For example:

  o The retraining frequency of the neural network is mentioned, but no explanation about the selection of the values is provided. Explaining why this frequency was chosen would help readers understand its impact.

*Thank you for pointing out the unclarity regarding IGM model choices. All our parameter choices are based on Leger et al. (2025), except for the higher retraining frequency for the finest-resolution simulations. With IGM's is physics-informed strategy, a higher retraining frequency permits to maintain fidelity level of the Blatter-Pattyn model. The retraining frequencies of 2 and 7 used in our experiments refer to retraining every 0.02-0.28 years (given the typical adaptive time step of 0.01-0.04 years). This means that the emulator is retrained based on the Blatter-Pattyn model multiple times a year, reducing deviations. For justifications of the retraining frequencies we refer to Jouvet and Cordonnier (2023) and Leger et al. (2025). We will add this information and references on the retraining parameters to the revised manuscript. Additionally, we will make sure that it is clear which model parameters are based on Leger et al. (2025) and which are defined by us.*

  o Elevation bands for low, mid, and high altitudes are defined as 0–1200 m, 1200–2400 m, and >2400 m. A brief justification or reference would improve clarity.

*Thank you for raising this point. We would like to clarify that the elevation bands are used to roughly distinguish between low, mid, and high altitudes across the Alp's hypsometry and elevation range. We use these elevation bands to distinguish between topographic features that are typical for these altitudes (e.g., deeply incised valleys at low altitudes). We propose to add the elevation ranges to panels b and c of Fig. 1 to visualize that the elevation bands distinguish between gentle-sloping regions at 0-1200 m a.s.l., areas with intermediate slopes at 1200-2400 m a.s.l., and steep regions above 2400 m a.s.l., e.g.,:*

[Figure]

- While the study explores a wide range of resolutions, all resampling is performed using cubic interpolation. Would the authors expect differences by using other techniques (bilinear, kriging…)? Could this lead to some misleading results?

*We will add a justification for the cubic resampling method in the revised manuscript, e.g.: "For all DEM resampling of input fields, we use cubic convolution due to its high accuracy in resolving topographic features such as slope angles compared with other analytical methods like bilinear or kriging resampling (Mihn et al., 2024)." Consequently, we expect slightly higher differences between fine and coarse resolution DEMs using other resampling methods, which might lead to even stronger resolution effects. However, we consider differences due to resampling methods to be minor compared to the overall resolution effects analyzed.*

*Please note that we used cubic resampling for the preparation of the input DEMs, while for post-processing comparisons of our model outputs at different resolutions, we instead used nearest neighbour resampling to preserve the pixel structure at coarse resolutions (lines 171-172).*

**Results**

- Some aspects of the results are difficult to interpret solely from the figures, particularly given the reliance on supplementary material. Consider highlighting the key patterns in the main text to improve readability.

*We fully agree that the readability of the results section should be improved. To that end (and in lines with comments from reviewer 1), we propose to reduce the number of figures (removing Figs. 6 and 9) and generally shorten the paragraphs to highlight main aspects. We believe that the key patterns are visible in the figures already provided in the main text and references to the supplementary material will be reduced substantially.*

- While the manuscript identifies three resolution "modes", the evidence is not entirely clear except for Figure 8 and not for all the outputs and elevation bands.

*Thank you for the comment. In line with the comments from reviewer 1, we agree to remove the term "modes" and focus on the comparison to the 50 m reference run. We would like to emphasize resolution effects at fine and coarse resolutions (e.g., generally thinner and faster-flowing ice at fine resolutions), instead of mainly focusing on strong changes at intermediate resolutions.*

*However, we argue that there is more evidence for the three resolution "modes" than Figure 8, e.g., the maximum ice volume can be divided into rather consistently low ice volume at fine resolutions and consistently high ice volume at coarse resolutions, with high increase in ice volume at intermediate resolutions (Fig. 3). Moreover, the ice thickness differences are small at fine resolutions and thickness differences strongly increase with resolution at intermediate resolutions (Fig. 5).*

**Discussion**

- The discussion is generally relevant but could be more explicitly connected to practical implications. For example, explaining how these findings could guide modelers in selecting appropriate spatial resolutions would increase the impact (other than just going for finer resolution as a general rule).

*Thank you for this comment. As mentioned above, we think it best to shift the focus from the "critical resolution mode" to a comparison to the 50 m reference run, which shows that model results at 300 m and finer are generally comparable to the reference. Therefore, we can conclude that a resolution of 300 m is sufficiently fine to accurately simulate glacier dynamics in our exemplary ice field in the Western Alps. Removing the confusing "resolution modes" and instead identifying a "sufficiently fine resolution" will already be more useful from a practical perspective. Our paper might therefore provide a valuable reference for determining a resolution for a mountain region similarly steep to the Western Alps.*

*Moreover, the experiments using a smoothed topography show similar resolution-effects that depend on altitude and are non-linear. From those experiments, we can conclude that a resolution of 400-500 m is sufficiently fine to accurately model the icefield in smoother terrain than the Western Alps. A discussion of different resolutions for ice-sheets (Greenland, Antarctica) in context of our experiments for the Western Alps will provide further guidance for modellers.*

*Because we find that input and output variables vary differently to resolution, it is very difficult to give specific recommendations for spatial resolution based on any DEM. However, we believe that a recommendation for a coarse limit of resolution can be based on an inherent wavelength of the landscape. For example, if the spatial resolution is coarser than the characteristic scale of the valleys, it is definitely too coarse. In our study, we do not consider this case because the*

*coarsest resolution of 2 km is fine enough to capture the main valley system. However, at even coarser resolution of, e.g., 5 or 10 km, more topographic characteristics would be lost, leading to even stronger resolution effects.*

- I feel that maybe more detailed exploration of this 400-800 "critical" mode should be addressed before to clearly talk about strong sensibility.

*We agree and as pointed out above, we will remove the "resolution modes". Strong changes at intermediate resolutions will be described more explicitly instead, which will add clarity to the results and discussion. Since we would like to focus more on differences compared to the 50 m reference run, we believe it to be more valuable to contrast ice field characteristics at fine (generally thinner glaciers and faster ice flow) and coarse resolutions – instead of focusing on intermediate resolutions.*

*We would like to add that the strong sensitivity at intermediate resolutions is visible in Figs. 8, 5, and 3, where generally small changes in resolution lead to quite high differences in output variables. To give an example for the higher sensitivity, the maximum ice volume from the 400 m run increases by over 25 % when using a 500 m resolution. For comparison, the difference of maximum ice volume is less than 1 % between the 50 and 100 m runs and ~1.5 % between the 1800 and 2000 m runs, which indicates that the intermediate resolutions are more sensitive to small changes in spatial resolutions.*

**Conclusions**

- I like the bullet structure, but I think that some points could be rearranged for more clarity.

*Thank you for this comment. We agree that the conclusion points could be clearer. By integrating previous comments and removing the phrase "resolution modes" from the manuscript, we expect to enhance clarity. Additionally, the points on mid and low altitudes can be combined and focused on resolution effects in the valleys (which is also true for the corresponding discussion paragraphs in section 4.1).*

*We admit that it might be confusing to start the conclusions with point (i) by contrasting ice field characteristics at fine and coarse resolutions, followed by a more detailed description of resolution effects at different bedrock altitudes later in points (iii)-(v). We decided on this structure because we ordered the points based on importance. However, we are happy to rearrange the points by, e.g., putting points (i) and (iii)-(v) closer together.*

**Figures and Supplementary Material**

- While the figures themselves are informative, the main text often relies on references to supplementary material to interpret the results. This can make the narrative difficult to follow. Consider ensuring that the key patterns and findings are clearly described within

the main text, using the supplementary figures as support rather than as primary evidence.

*Thank you for the suggestion. The references to supplementary figures are mainly additional information. We are confident that we are able to present our results in a more concise way in the revised manuscript without excessive reference to supplementary figures for details that we won't show in the main text.*

**Minor Comments**

Here I will try to skip the comments raised from the other reviewer so we don't need to repeat the same things.

**L108:** Please, justify this choice of (0.01-0.04).

*The time step in IGM is adaptive and not static, i.e., it is not a parameter chosen by the user but instead calculated in IGM based on the CFL condition that ensures numerical stability for the ice thickness evolution. We propose to remove the values for a typical time step and retraining time step frequencies from the manuscript and instead give the retraining frequencies in years: For adjustments to new ice field states attained through run time, we retrained the physics-informed neural network multiple times within a single model year (every ~0.18 years). The on-the-run retraining in the highest-resolution simulations at 50 and 100 m was even performed more often (every ~0.05 years)."*

**L178:** Please, justify the selection of the elevation bands.

*As mentioned above, we agree to add an explanation for the choice of elevation bands, e.g.: "We used elevation bands of 0–1200, 1200–2400, and >2400 m a.s.l. to distinguish between low, mid, and high altitudes across the Alp's elevation range (Fig. 1a) that feature deeply incised, gently-sloping valleys at low altitudes and steep mountain summits at high altitudes." We propose to add the elevation ranges to panels b and c of Fig. 1 to visualize that the elevation bands distinguish between gentle-sloping regions at 0-1200 m a.s.l., areas with intermediate slopes at 1200-2400 m a.s.l., and steep regions above 2400 m a.s.l. (see updated Fig. 1 above).*

**L184:** effects -> affects?

*Yes, we will adopt this in the revised manuscript.*

**L209:** This 10 times more is not clear to me, at least from the cited figures.

*Thank you for highlighting the need for clarification. Due to the y-axis, it is not really visible in the figure that the 2000 m simulation has 10 times more ice volume than the 50 m run at 1000 years of model time. However, it is clear in the figure that at 1000 years, the coarse resolutions have (relatively speaking) much more ice volume than the fine resolutions. In numbers, the 50 m*

*run has ~12 km² ice volume at 1000 years, while the 2 km run has ~160 km² ice volume at this time step. These numbers will be included in the text.*

**L254:** It is not clear to me looking at Fig. S12.

*Thank you for pointing this out. We apologize for the typo and refer to Fig. S3 that shows the spatial distribution of ice thickness at full glaciation.*

**L426-7:** I really don't see those greatest relative changes at high altitudes. Probably you have the numbers, but looking at Fig. 8f such a thing is not clear to me.

*Thank you for the comment. As mentioned above, the analysis of the Arrhenius factor does not add many insights, and we would like to remove this from our analysis to focus more on key variables (thickness, velocity, temperature). We would like to add that the mean Arrhenius factor at high altitudes is ~66 M $Pa^{-3}$ $yr^{-1}$ for the 50 m run and ~112 $Pa^{-3}$ $yr^{-1}$ for the 2 km run, leading to a difference of almost 60 % relative to the 50 m run.*

**L429-31:** Again, this is not obvious to me just looking at Fig. 8.

*Similar to the previous comment, we propose to remove the Arrhenius factor parts from the manuscript.*

**L474-8:** Nothing to object here, the opposite, really clear statement and good point resulting from your work.

*Thank you very much.*

**L500:** Here you suggest to go to finer than km- and ideally 500-m scale- but your defined critical mode ranges from 400 to 800m. I found this a little bit contradictory.

*Thank you for spotting this inconsistency, this should be changed to "400-m-scale".*

**Overall Recommendation**

Even if this is a relevant and well-executed study that addresses an important question in glacier modelling, especially in regional or global studies, I consider the manuscript as major revisions after considering some points:

1. Clarifying model limitations and generalization potential.

2. Justifying some methodological choices, such as elevation bands, retraining frequency, and resampling method.

3. Streamlining results and discussion to highlight the most important findings and improve readability. Exploring a bit more this critical mode sensibility.

4. Enhancing figure clarity and reducing overreliance on supplementary material.

I believe that addressing these points will help the resulting paper become a solid and valuable contribution to the field.

*Thanks for the valuable suggestions, which we hope we have addressed in the above mentioned planned modifications.*

References

Aschwanden, A., Fahnestock, M. A., and Truffer, M.: Complex Greenland outlet glacier flow captured, Nat Commun, 7, 10524, https://doi.org/10.1038/ncomms10524, 2016.

Cuzzone, J. K., Schlegel, N.-J., Morlighem, M., Larour, E., Briner, J. P., Seroussi, H., and Caron, L.: The impact of model resolution on the simulated Holocene retreat of the southwestern Greenland ice sheet using the Ice Sheet System Model (ISSM), The Cryosphere, 13, 879–893, https://doi.org/10.5194/tc-13-879-2019, 2019.

Jouvet, G. and Cordonnier, G.: Ice-flow model emulator based on physics-informed deep learning, J. Glaciol., 69, 1941–1955, https://doi.org/10.1017/jog.2023.73, 2023.

Jouvet, G., Cohen, D., Russo, E., Buzan, J., Raible, C. C., Haeberli, W., Kamleitner, S., Ivy-Ochs, S., Imhof, M. A., Becker, J. K., Landgraf, A., and Fischer, U. H.: Coupled climate-glacier modelling of the last glaciation in the Alps, J. Glaciol., 69, 1956–1970, https://doi.org/10.1017/jog.2023.74, 2023.

Leger, T. P. M., Jouvet, G., Kamleitner, S., Mey, J., Herman, F., Finley, B. D., Ivy-Ochs, S., Vieli, A., Henz, A., and Nussbaumer, S. U.: A data-consistent model of the last glaciation in the Alps achieved with physics-driven AI, Nat Commun, 16, 848, https://doi.org/10.1038/s41467-025-56168-3, 2025.

Pattyn, F., Perichon, L., Aschwanden, A., Breuer, B., de Smedt, B., Gagliardini, O., Gudmundsson, G. H., Hindmarsh, R. C. A., Hubbard, A., Johnson, J. V., Kleiner, T., Konovalov, Y., Martin, C., Payne, A. J., Pollard, D., and Price, S.: Benchmark experiments for higher-order and full-Stokes ice sheet models (ISMIP–HOM), The Cryosphere, 2008.

Rückamp, M., Goelzer, H., and Humbert, A.: Sensitivity of Greenland ice sheet projections to spatial resolution in higher-order simulations: the Alfred Wegener Institute (AWI) contribution to ISMIP6 Greenland using the Ice-sheet and Sea-level System Model (ISSM), The Cryosphere, 14, 3309–3327, https://doi.org/10.5194/tc-14-3309-2020, 2020.

Rückamp, M., Kleiner, T., and Humbert, A.: Comparison of ice dynamics using full-Stokes and Blatter–Pattyn approximation: application to the Northeast Greenland Ice Stream, The Cryosphere, 16, 1675–1696, https://doi.org/10.5194/tc-16-1675-2022, 2022.

Williams, C. R., Thodoroff, P., Arthern, R. J., Byrne, J., Hosking, J. S., Kaiser, M., Lawrence, N. D., and Kazlauskaite, I.: Calculations of extreme sea level rise scenarios are strongly dependent on ice sheet model resolution, Commun Earth Environ, 6, 60, https://doi.org/10.1038/s43247-025-02010-z, 2025.

Winkelmann, R., Martin, M. A., Haseloff, M., Albrecht, T., Bueler, E., Khroulev, C., and Levermann, A.: The Potsdam Parallel Ice Sheet Model (PISM-PIK) – Part 1: Model description, The Cryosphere, 5, 715–726, https://doi.org/10.5194/tc-5-715-2011, 2011.